# How Does the Lagrangian Guide Safe Reinforcement Learning through Diffusion Models?

**Xiaoyuan Cheng** [* 1]  **Wenxuan Yuan** [* 2]  **Boyang Li** [3]  **Yuanchao Xu** [4]  **Yiming Yang** [1]  **Hao Liang** [5]  **Bei Peng** [6]
**Robert Loftin** [6]  **Zhuo Sun** [7]  **Yukun Hu** [1]

∗ *Indicates the Equal Contributing Authors*

## Abstract

Diffusion policy sampling enables reinforcement learning (RL) to represent multimodal action distributions beyond suboptimal unimodal Gaussian policies. However, existing diffusion-based RL methods primarily focus on offline setting for reward maximization, with limited consideration of safety in online settings. To address this gap, we propose **Augmented Lagrangian-Guided Diffusion** (**ALGD**), a novel algorithm for off-policy safe RL. By revisiting optimization theory and energy-based modeling, we show that the instability of primal–dual methods arises from the non-convex Lagrangian landscape. In diffusion-based safe RL, the Lagrangian can be interpreted as an energy function guiding the denoising dynamics; counter-intuitively, direct usage destabilizes both policy generation and training. ALGD resolves this issue by introducing an augmented Lagrangian that locally convexifies the energy landscape, yielding a stabilized policy generation and training, without altering the distribution of optimal policy. Theoretical analysis and extensive experiments demonstrate that ALGD is both theoretically grounded and empirically effective, achieving strong and stable performance across diverse environments. See project page: `https://github.com/Wenxuan52/ALGD`.

[1]Dynamic Systems Lab, University College London [2]College of Computing and Data Science, Nanyang Technological University [3]Department of Mechanical and Aerospace Engineering, University of California, San Diego [4]Graduate School of Science, Kyoto University [5]Department of Informatics, King's College London [6]School of Computer Science, University of Sheffield [7]School of Statistics and Data Science, Shanghai University of Finance and Economics. Correspondence to: Xiaoyuan Cheng, Wenxuan Yuan <`ucesxc4@ucl.ac.uk`, `YUAN0186@e.ntu.edu.sg`>, Yukun Hu <`yukun.hu@ucl.ac.uk`>.

*Proceedings of the $43^{rd}$ International Conference on Machine Learning*, Seoul, South Korea. PMLR 306, 2026. Copyright 2026 by the author(s).

## 1. Introduction

Reinforcement learning (RL) has achieved tremendous successes in various domains (Silver et al., 2016; Ibarz et al., 2021; Guo et al., 2025). In many practical situations, direct interaction with the environment is expensive and potentially hazardous (García & Fernández, 2015; Gu et al., 2024). To address the safety issues, safe RL incorporates safety considerations by enforcing constraints that keep certain risk measures below predefined thresholds (Achiam et al., 2017), while optimizing the expected reward. Recently, safe RL has gained increasing attention as a crucial step toward deploying RL in real-world applications (Dulac-Arnold et al., 2021). From an optimization perspective (Zheng et al., 2024), existing safe RL methods can be broadly categorized into *primal–dual* and *hard-constrained* methods.

**Primal-Dual Methods.** The first class, primal–dual methods, enforces safety in expectation by introducing Lagrange multipliers to relax the constraints, enabling the agent to balance reward maximization and constraint satisfaction during training (Chow et al., 2018a; Tessler et al., 2018; Ding et al., 2020; Sootla et al., 2022; Yang et al., 2022; Liu et al., 2022b; 2023). In general, these methods alternate between the primal update for policy optimization and the dual update for Lagrange multiplier adjustment. While they provide solid theoretical guarantees for converging to constraint-satisfying policies (Achiam et al., 2017; Zhang et al., 2020; Altman, 2021), they often exhibit severe instability in practice (So & Fan, 2023; Zhang et al., 2025b). This instability primarily arises from the tight coupling between cost estimation and policy optimization during online exploration. In particular, during the early stages of training, inaccurate cost estimates can drive updates of a unimodal Gaussian policy toward suboptimal regions. Due to its limited expressive capacity, such a policy is unable to move beyond these reinforced suboptimal modes, leading to biased policy updates and unstable dual variable updating. This issue is further exacerbated in *off-policy* settings (Wu et al., 2024), where distributional shift amplifies bias and variance in cost estimation, resulting in oscillatory dual variables and unstable policy optimization. As a consequence,

*Table 1.* Comparison of representative safe RL methods and our method. Primal-dual methods, typically on-policy in online training, suffer from instability due to oscillating dual updates. Hard-constrained approaches require accurate exploration of the maximal safe set, which is often difficult and conservative. Most diffusion-based RL methods are limited to offline settings and poorly explore safety constraints in online settings. Our **ALGD** falls within the primal-dual family but enables online safe RL (*off-policy*) by revisiting an energy-based optimization perspective, bridging the gap between expressive generative policies and stable constrained optimization.

| Methods | Online | Conservative | Reward | Training Stability | Policy Distribution |
|---|---|---|---|---|---|
| Primal-Dual | ✓ | ✗ | High | ✗ | Gaussian |
| Hard-Constraint | ✓ | ✓ | Low | ✓ | Deterministic/Gaussian |
| Diffusion-Based | ✗ | – | High | ✓ | Multimodal |
| **ALGD (ours)** | ✓ | ✗ | **High** | ✓ | **Multimodal** |

most existing primal-dual methods are on-policy by design and require a large number of interaction samples to converge, which is problematic in safety-critical applications where environment interactions may be inherently risky.

**Hard-Constrained Methods.** To mitigate such instability in primal-dual methods, various hard-constrained approaches have been proposed, including Lyapunov-based methods (Chow et al., 2018b; Zhang & Fan, 2024), control barrier functions (Qin et al., 2022; Ma et al., 2021), Hamilton–Jacobi (HJ) reachability analysis (Ganai et al., 2023; 2024), safety shielding (Wagener et al., 2021), and projection-based techniques (Lin et al., 2024). Unlike primal–dual methods that enforce safety in expectation, hard-constrained methods theoretically guarantee state-wise constraint satisfaction during execution. Specifically, these methods usually define a sublevel set of the constraint function as a safe set (also referred to as a control-invariant set in control theory) (Yu et al., 2022), which ensures that the agent's behavior remains within this region (So & Fan, 2023; Qin et al., 2024; Zhang et al., 2025b;c; Ganai et al., 2024). However, since the largest safe (or control-invariant) set is usually unknown (Choi et al., 2025), online RL with parameterized policies (e.g., Gaussian or deterministic) may fail to approximate it accurately. Consequently, conservative policies derived from overly restricted safe sets tend to limit exploration and reduce achievable rewards.

**Diffusion-based RL.** Beyond the convex optimization perspective, an emerging line of research explores reinforcement learning through diffusion models, as the generated policies can represent distributions beyond the Gaussian assumption (Janner et al., 2022; Wang et al., 2022; 2024). However, because the generative nature of diffusion policies relies on fixed offline data distributions (Kang et al., 2023; Uehara et al., 2024; Park et al., 2025), most existing diffusion-based approaches remain confined to the offline RL setting. Recent studies have further investigated safe offline policy generation using diffusion models (Xiao et al., 2023; Cheng et al., 2025), primarily focusing on learning policies from static datasets. While a few works have begun exploring the connection between online RL and diffusion models (Ding et al., 2024; Psenka et al., 2023; Ma et al.,

2025), safety considerations remain largely unexplored in such settings (see a more detailed review in Appendix A).

In light of the limitations of existing approaches (see Table 1), we turn to diffusion models for their superior expressiveness. However, the key challenge lies in how to seamlessly introduce safety constraints into the diffusion policy during online exploration without compromising training stability and optimality. In this work, we propose Augmented Lagrangian-Guided Diffusion (ALGD), a principled framework that reformulates safe RL as guided diffusion. Our contributions are threefold: (1) We provide a novel reformulation of safe RL in the diffusion framework by interpreting the Lagrangian objective as the energy function governing the reverse diffusion process. Through this formulation, we reveal a counter-intuitive but fundamental limitation: directly applying the standard Lagrangian induces a highly nonconvex energy landscape, resulting in unstable score fields and inconsistent Boltzmann distributions. (2) We theoretically show how this issue can be resolved via an augmented Lagrangian formulation that locally convexifies the energy landscape. We prove that this modification stabilizes the diffusion dynamics without altering the optimal policy distribution of the original constrained problem. (3) Based on this insight, we develop a novel algorithm and provide a discrepancy analysis that formally characterizes the gap between the learned diffusion policy and the ideal constrained solution, demonstrating the feasibility of our approach. Empirically, ALGD achieves competitive returns while consistently reducing constraint violations compared to prior safe RL methods.

## 2. Preliminaries

Following the notation in Psenka et al. (2023), we start from a general nonlinear controlled stochastic dynamics with a continuous action space. In this section, we first introduce key notations, followed by the formal problem formulation.

**Notation.** We denote the environment time step by $t$. The state and action spaces are compact sets $\mathcal{S} \subseteq \mathbb{R}^n$ and $\mathcal{A} \subseteq \mathbb{R}^m$, respectively. Following standard notation in stochastic calculus, $ds$ denotes an infinitesimal increment

of a variable $s$, and $\nabla f$ denotes the gradient of a function $f$. The reward function is $r : \mathcal{S} \times \mathcal{A} \to \mathbb{R}$, and the cost function is $c : \mathcal{S} \to \mathbb{R}_{\geq 0}$. The policy $\pi(a|s)$ defines a conditional distribution over actions given state $s$. The initial state $s_0$ is drawn from distribution $d_0$. We use $\gamma \in [0, 1]$ to denote the discount factor and $\beta > 0$ the temperature parameter for policy parametrization. Throughout the paper, superscripts (e.g., $a^\tau$) index diffusion steps, while subscripts (e.g., $s_t, a_t$) denote environment time steps.

**Diffusion Model.** In this work, we consider the variance exploding (VE) stochastic differential equation (SDE) (Song et al., 2020) in a continuous action space, which can be written as (Akhound-Sadegh et al., 2024)

$$da^\tau = \sqrt{\frac{d\sigma^2(\tau)}{d\tau}} \, dB^\tau,$$

where $B^\tau$ denotes the standard Brownian motion and $\sigma(\tau)$ is a monotonically increasing noise scale. As time progresses, the prior policy distribution $\pi^0(a|s)$ is progressively convolved with Gaussian noise of increasing variance $\sigma^2(\tau)$, yielding the marginal distribution

$$\pi^\tau(a^\tau|s) = \int \pi^0(a^0|s) \mathcal{N}\left(a^\tau; a^0, \sigma^2(\tau)I\right) da^0, \quad (1)$$

which smoothly transitions from the target distribution $\pi(a|s)$ to an isotropic Gaussian distribution as $\tau$ increases. The VE SDE thus describes a diffusion process where the variance of $a^\tau$ increases over time, gradually transforming the data distribution into an isotropic Gaussian.

In the reverse-time formulation, the dynamics follow

$$da^\tau = \left[ -\frac{d\sigma^2(\tau)}{d\tau} \phi(s, a^\tau, \tau) \right] d\tau + \sqrt{\frac{d\sigma^2(\tau)}{d\tau}} \, dB^\tau, \quad (2)$$

where $\phi(s, a^\tau, \tau)$ approximates the score function $\nabla_a \log \pi^\tau(a^\tau|s)$ for denoising and recovering the clean data distribution (Song et al., 2020).

**Problem Formulation.** The objective of safe RL is to optimize expected reward while ensuring that the accumulated cost remains below a specified threshold $h$ in this case. Formally, the problem is:

$$\begin{aligned} \max_\pi \quad & \mathbb{E}_{s_0 \sim d_0}[V^\pi(s_0)] \\ \text{s.t.} \quad & \mathbb{E}_{s_0 \sim d_0}[V_c^\pi(s_0)] \leq h, \end{aligned} \quad (3)$$

where $V^\pi(s_0) = \mathbb{E}_{a_0 \sim \pi(\cdot|s_0)}[Q^\pi(s_0, a_0)] := \mathbb{E}_\pi\left[\sum_{t=0}^\infty \gamma^t r(s_t, a_t)|s_0\right]$ is the value function, and $V_c^\pi(s_0) = \mathbb{E}_{a_0 \sim \pi(\cdot|s_0)}[Q_c^\pi(s_0, a_0)] := \mathbb{E}_\pi\left[\sum_{t=0}^\infty \gamma^t c(s_t)|s_0\right]$ is the cost value function. We define the standard Lagrangian (hereafter abbreviated as Lagrangian) at the state–action level as:

$$\mathcal{L}(s_0, a_0, \lambda) := -Q^\pi(s_0, a_0) + \lambda\left(Q_c^\pi(s_0, a_0) - h\right),$$

where $\lambda \geq 0$ is the Lagrangian multiplier. The optimization of (3) is carried out

$$\max_{\lambda \geq 0} \min_\pi \mathbb{E}_{\substack{s_0 \sim d_0, \\ a_0 \sim \pi(\cdot|s_0)}} [\mathcal{L}(s_0, a_0, \lambda)]. \quad (4)$$

Here, we assume that the constrained problem is feasible. In practice, solving (4) over continuous action spaces commonly adopts entropy-regularized primal-dual optimization, yielding a stochastic policy distribution. Under this formulation, three main limitations arise in primal-dual methods.

(L1) In entropy-regularized policy optimization (see Appendix C.1), the policy is parameterized as a *Boltzmann distribution* with the Lagrangian $\mathcal{L}$ serving as the energy:

$$\pi(a|s) = \frac{\exp\left(-\frac{\mathcal{L}(s,a,\lambda)}{\beta}\right)}{Z}, \quad (5)$$

where $\beta > 0$ is the temperature factor and $Z = \int_a \exp\left(-\frac{\mathcal{L}(s,a,\lambda)}{\beta}\right) da$ is the (intractable) partition function. In previous safe RL literature, this Boltzmann policy distribution is often approximated as a Gaussian distribution for tractability, which deviates from the true one and introduces additional bias and a mismatch between theory and implementation.

(L2) The dual variable $\lambda$ is updated using inaccurate cost value estimation evaluated under a misspecified policy, resulting in unstable dual updates due to a mismatch with the true Boltzmann distribution.

(L3) Since the Lagrangian $\mathcal{L}(s, a, \lambda)$ in (5) directly shapes the energy landscape, oscillations of $\lambda$ induce large fluctuations in Boltzmann distribution, leading to instability.

We pose the following research questions:

> *Are diffusion models capable of addressing the key limitations in safe RL, and what designs are required to overcome the challenges that arise from their integration?*

To explore this, we first investigate the fundamental connection between diffusion models and safe RL, then analyze the key difficulties that arise when combining them, and finally present how to overcome those challenges to benefit safe RL both theoretically and algorithmically.

## 3. Method

We first uncover the connections between diffusion models and safe RL and their challenges. We then formulate the Augmented Lagrangian-Guided Diffusion (ALGD) framework to resolve these issues, and finally introduce a practical algorithm based on ALGD for online safe RL.

### 3.1. When Diffusion Meets Safe RL

Firstly, we reformulate the policy gradient of the safe RL (4) as

$$\mathbb{E}_{\substack{s_0 \sim d_0 \\ a_0 \sim \pi_\theta}} \left[ -\sum_{t=0}^{\infty} \mathcal{L}(s_t, a_t, \lambda) \big( \underbrace{\sum_{\tau=K}^{1} \nabla_\theta \log \pi_\theta^\tau(a_t^{\tau-1}|a_t^\tau, s_t)}_{\text{policy gradient with refinement}} \big) \right].$$

(6)

where each intermediate policy $\pi_\theta^{\tau-1}(a_t^{\tau-1}|a_t^\tau, s_t)$ refines the previous step along a reverse diffusion chain. The resulting marginal distribution of the final action can be written as

$$\pi_\theta^0(a_t^0|s_t) = \int_{a_t^{1:K}} \pi_\theta^K(a_t^K|s_t) \prod_{\tau=K}^{1} \pi_\theta^{\tau-1}(a_t^{\tau-1}|a_t^\tau, s_t) da_t^{1:K},$$

(7)

which represents a multi-step refinement process that transforms a Gaussian prior $\pi_\theta^K \sim \mathcal{N}(0, \sigma^2(K)I)$ into the policy distribution $\pi_\theta^0$ shown in (5). This iterative refinement is a discrete approximation of the reverse-time SDE defined in (2), where the drift term corresponds to $-\frac{d\sigma^2(\tau)}{d\tau} \phi_\theta(s, a^\tau, \tau)$. Under this view, updating the policy gradient in parameter space is equivalent to performing *score matching* with respect to the score function of the reverse-time SDE in (2). This enforces the learned diffusion score field $\phi_\theta(s, a^\tau, \tau)$ to align with the instantaneous score $\nabla_a \log \pi^\tau(a^\tau|s)$ that drives the reverse diffusion process. Note that the score network shares the same parameter $\theta$ as the policy, since the entire multi-step refinement process is governed by $\phi_\theta(s, a^\tau, \tau)$. When this score field is accurately learned, the resulting distribution $\pi_\theta^0(a|s)$ converges to the intended Boltzmann distribution (5).

Generally, the intermediate score $\nabla_a \log \pi^\tau(a^\tau|s)$ is intractable to compute, as the intermediate policy distribution $\pi^\tau$ lacks a closed-form expression during the reverse diffusion process. *Since the score estimation in Psenka et al. (2023) is misspecified except $\tau = 0$ (Dong et al., 2025), we propose to calculate the exact score function based on the Lagrangian $\mathcal{L}(s, a, \lambda)$.*

**Proposition 3.1** (Lagrangian-guided score function). *Consider the VE SDE with the reverse-time formulation in (2), the Lagrangian $\mathcal{L}(s, a, \lambda)$ is globally Lipschitz continuous over the entire state-action spaces $\mathcal{S}$ and $\mathcal{A}$. By fixing the Lagrange multiplier $\lambda$ and sampling trajectories $(s_t, a_t)$ starting from $(s_0, a_0)$, the optimization objective in (4) attains its optimal value when the score field $\phi_*(s, a^\tau, \tau)$ equals*

$$\mathbb{E}_{a^{0|\tau}} \left[ - \frac{\exp\left( -\frac{\mathcal{L}(s, a^{0|\tau}, \lambda)}{\beta} \right)}{\mathbb{E}_{a^{0|\tau}} \left[ \exp\left( -\frac{\mathcal{L}(s, a^{0|\tau}, \lambda)}{\beta} \right) \right]} \frac{\nabla_a \mathcal{L}(s, a^{0|\tau}, \lambda)}{\beta} \right],$$ (8)

*with the posterior distribution $a^{0|\tau} \sim \mathcal{N}(a^\tau, \sigma^2(\tau)I)$ and the first part is the normalized weight. Therefore, the op-*

*timal score function $\phi_*$ aligns with the Lagrangian gradient in the action space (see proof and discussion in Appendix C.1).*

> **Why Lagrangian-guided diffusion is problematic?**
>
> Proposition 3.1 implies that, the optimal score field $\phi_\theta(s, a, \tau)$ is entirely shaped by the Lagrangian $\mathcal{L}(s, a, \lambda)$. However, in practice, two major challenges arise from the gradient term $\nabla_a \mathcal{L}(s, a, \lambda)$ in (8).
>
> First, in the early stages of training, the Lagrangian gradient $\nabla_a \mathcal{L}(s, a, \lambda) = -\nabla_a Q^\pi(s, a) + \lambda \nabla_a Q_c^\pi(s, a)$ is computed via the auto-differentiation of $Q^\pi$ and $Q_c^\pi$. Since estimators are usually highly nonconvex, the resulting gradient field tends to be noisy and irregular, introducing many suboptimal directions in the score field. Thus, the diffusion process may sample from unstable or low-reward regions, making the target Boltzmann distribution difficult to approximate.
>
> Second, the Lagrange multiplier $\lambda$ is typically initialized with a small value and updated during learning. When the cost estimate of $Q_c^\pi$ is inaccurate (Thrun & Schwartz, 2014), the update of $\lambda$ may fluctuate or drift, leading to a distorted energy landscape $\mathcal{L}(s, a, \lambda)$ and a mismatched score field. In this case, the policy can continue to sample from high-reward yet high-cost regions, as the weak penalty term fails to effectively enforce the constraint. This results in a distributional mismatch between the learned policy distribution and intended safe Boltzmann distribution.

Overall, these phenomena indicate that the limitations identified in (L2) and (L3) are not resolved by applying Lagrangian-guided diffusion as Proposition 3.1 directly. Instead, the instability of the Lagrangian gradient and the dual variable is directly inherited and even amplified under the diffusion formulation. A straightforward visualization of energy landscape induced by the Lagrangian is in Figure 1.

### 3.2. Locally Convexified Energy Landscape

To overcome these limitations, we propose the *augmented Lagrangian* to guide diffusion process. Following the classical results in constrained optimization (Hestenes, 1969; Powell, 1969; Rockafellar, 1976), we obtain the augmented Lagrangian $\mathcal{L}_A$ corresponding to the problem (4) as

$$-Q^\pi(s, a) + \frac{\left( \left[ \lambda + \rho \big( Q_c^\pi(s, a) - h \big) \right]_+^2 - \lambda^2 \right)}{2\rho}, \quad (9)$$

where $\rho > 0$ controls the magnitude of the quadratic penalty and thus the degree of local convexification, $[\Box]_+ = \max(0, \Box)$ enforces non-negativity of the dual variables. The quadratic term *locally increases curvature around constraint boundaries*, yielding a smoother and partially convex

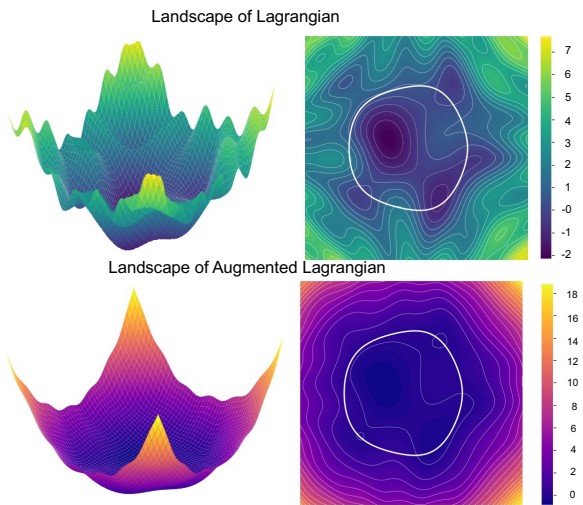

Landscape of Lagrangian

Landscape of Augmented Lagrangian

*Figure 1.* Visualization of energy landscapes for a differential-drive mobile robot (Contreras et al., 2017) after 100 training episodes based on our methods (see the two algorithms implementation in Appendix D). **Top:** The landscape of standard Lagrangian induces a highly irregular and non-convex energy surface with sharp curvature, reflecting unstable denoising dynamics. **Bottom:** The landscape of augmented Lagrangian yields a smoother and locally convexified energy landscape, effectively regularizing the score field. In both contour plots, the *white circle* indicates the safe policy region, within which policies satisfy the safety constraints.

energy landscape that mitigates gradient irregularities and stabilizes the denoising dynamics.

**Theorem 3.2.** *Under mild regularity assumptions on $Q^\pi$ and $Q_c^\pi$ (i.e., boundedness and smoothness), $\mathcal{L}_A(s, a, \lambda)$ in (9) satisfies the following properties:*

*(a) (**Local landscape convexification**). The quadratic penalty term in $\mathcal{L}_A$ introduces additional positive semidefinite curvature in a neighborhood of the feasible region, resulting in smoother gradients and a better-conditioned local energy landscape for denoising dynamics. In particular,*

$$\nabla_a^2 \mathcal{L}_A(s, a, \lambda) = \nabla_a^2 \mathcal{L}(s, a, \lambda) + \boxed{\rho \, \nabla_a Q_c^\pi(s, a) \nabla_a Q_c^{\pi\top}(s, a)} + \mathcal{O}(|Q_c^\pi(s, a) - h|), \tag{10}$$

*where the dominant additional term is positive semidefinite.*

*(b) (**Invariant optimal policy and objective**). Fix the optimal dual variable $\lambda^*$, the augmented Lagrangian preserves both the optimal value and the optimal policy distribution. Specifically, if $(\pi^*, \lambda^*)$ is a primal-dual optimal solution, then*

$$\min_\pi \mathbb{E}_{a\sim\pi}\big[\mathcal{L}_A(s, a, \lambda^*)\big] = \min_\pi \mathbb{E}_{a\sim\pi}\big[\mathcal{L}(s, a, \lambda^*)\big],$$

*and the induced Boltzmann distribution remains unchanged:*

$$\pi_A^*(a|s) = \pi^*(a|s).$$

*since* $\exp\left(-\frac{\mathcal{L}_A(s,a,\lambda^*)}{\beta}\right) = \exp\left(-\frac{\mathcal{L}(s,a,\lambda^*)}{\beta}\right)$. *Therefore, introducing the augmented form only reshapes the energy landscape without altering the optimal policy (i.e., the resulting Boltzmann distribution in* (8)*). See proof in Appendix C.2.*

---

**Benefits of the augmented Lagrangian for diffusion.**

According to Theorem 3.2, the proposed augmented formulation directly addresses the two major limitations of the standard Lagrangian. First, by introducing the quadratic penalty term, $\mathcal{L}_A$ smooths out irregularities in the Lagrangian gradient $\nabla_a \mathcal{L}(s, a, \lambda)$ and adds a positive semidefinite curvature correction term $\rho \, \nabla_a Q_c^\pi \nabla_a Q_c^{\pi\top}$ (see (10)). This induces local convexity in the action space around the *constraint-active region*, where gradient instability is most pronounced. Such regions are critical during sampling, as local convexification around the constraint boundary reshapes the energy landscape so that gradients steer diffusion trajectories toward the feasible action set, reducing the likelihood of sampling unsafe actions. By regularizing the local curvature in these regions, the augmented formulation mitigates gradient irregularities and stabilizes the denoising dynamics. Second, unlike the standard Lagrangian where the dual variable $\lambda$ linearly penalizes constraint violations, the augmented term in (9) explicitly reshapes the energy landscape itself. This quadratic correction increases the energy in the unsafe action set $\{a \mid Q_c^\pi(s, a) > h\}$ and creates a smooth potential well around the feasible action set. As a result, when $\rho$ is large, the induced Boltzmann distribution $\pi_A(a|s) \propto \exp\left(-\frac{\mathcal{L}_A(s,a,\lambda)}{\beta}\right)$ concentrates its probability mass within safety-compliant regions, allowing the diffusion process to sample more reliably from stable, low-cost regions, even in the early stage of learning. Visualization of the energy landscape of the augmented Lagrangian is provided in Figure 1. Together, these two properties directly address limitations (L2) and (L3).

---

Our analysis shows that the augmented Lagrangian encourages policy samples to concentrate within the safe region, which in turn stabilizes the dual variable updates and mitigates oscillatory behavior. Moreover, since the augmentation does not alter the stationary conditions of the original constrained problem, both the optimal policy and the objective value remain invariant at convergence (see Theorem 3.2(b)). Figure 2 compares diffusion policies guided by the standard Lagrangian and the augmented Lagrangian. The results corroborate the improved denoising dynamics and enhanced training stability predicted by Theorem 3.2.

## 3.3. Practical Algorithm and Discrepancy Analysis

Next, we introduce our practical algorithm, Augmented Lagrangian-Guided Diffusion (ALGD), which illustrates how the augmented Lagrangian guides the diffusion process toward the intended Boltzmann distribution. Algorithm 1 in Appendix D summarizes the core idea of ALGD, which consists of three key components: (i) policy generation via reverse-time diffusion sampling by (2), (ii) critic learning for both $Q$ and $Q_c$ estimation, and (iii) score matching guided by the augmented Lagrangian. In particular, we highlight two aspects of the method: the ensemble learning of cost critics to improve constraint estimation, and the Monte Carlo estimation of the score function guided by the augmented Lagrangian.

**Ensemble Cost Critics.** To enhance the accuracy of cost-value estimation, ALGD employs an ensemble of $M$ cost critics $\{Q^c_{\psi_i}\}_{i=1}^M$. Each critic shares the same neural network but is initialized with randomized weights. All critics are trained independently on the same replay buffer using identical temporal-difference objectives. The ensemble outputs are then averaged to produce a single cost estimate $\bar{Q}_c(s,a)$ that is used in the augmented Lagrangian $\mathcal{L}_A(s,a,\lambda)$. This ensemble design improves the accuracy of the cost-value estimation, which in turn yields improved gradient estimation.

**Monte Carlo Score Estimation.** As shown in Proposition 3.1, the score function $\phi_A(s, a^\tau, \tau)$ follows the gradient of the Lagrangian $\mathcal{L}_A(s, a^{0|\tau}, \lambda)$ under a weighted expectation (red shadow of (8)). Since this expectation cannot be computed analytically, we approximate it using Monte Carlo sampling. ALGD employs a weighted Monte Carlo estimator based on a proposal distribution $q(a^{0|\tau}|s)$, chosen as a Gaussian $\mathcal{N}(a^\tau, \sigma^2(\tau)I)$ according to (8). The asymptotic unbiased expectation can thus be approximated using $N$ Monte Carlo samples $\{a^{0|\tau,(i)}\}_{i=1}^N$ drawn from $q(a^{0|\tau}|s)$:

$$\phi_A(s, a^\tau, \tau) \approx \sum_{i=1}^N - w_i \, \nabla_a \frac{\mathcal{L}_A(s, a^{0|\tau,(i)}, \lambda)}{\beta} \qquad (11)$$

with $\quad w_i = \dfrac{\exp\left[-\frac{1}{\beta}\mathcal{L}_A(s, a^{0|\tau,(i)}, \lambda)\right]}{\sum_{j=1}^N \exp\left[-\frac{1}{\beta}\mathcal{L}_A(s, a^{0,(j)}, \lambda)\right]} \quad$ and for all

$a^{0|\tau,(i)} \sim q(a^{0|\tau}|s)$ (see analysis in Lemma C.1). This weighted Monte Carlo estimation naturally incorporates the augmented Lagrangian energy into the score evaluation, assigning higher importance to low-energy actions (i.e., actions that better satisfy constraints and yield higher rewards). In practice, this reparameterization trick provides a differentiable surrogate for the exact score, allowing the diffusion model to denoise actions along the gradients of $\mathcal{L}_A(s, a, \lambda)$. The computationally intensive Monte Carlo

estimation is only used as score network training. At test time, the score network provides an amortized solution for efficient policy generation, see details in Algorithm 1.

**Theorem 3.3.** *Suppose that* $\pi^0_A(a^0|s) \propto \exp\left(-\frac{\mathcal{L}_A(s, a^{0|\tau}, \lambda)}{\beta}\right)$ *and its gradient* $\|\nabla_a \exp\left(-\frac{\mathcal{L}_A(s, a^{0|\tau}, \lambda)}{\beta}\right)\|$ *is sub-Gaussian. Then, with probability at least* $1 - 2\delta$*, the discrepancy between the generated policy distribution and the target Boltzmann distribution is bounded by*

$$D_{KL}\big(\pi^*(a|s) \,\|\, \tilde{\pi}^0_A(a^0|s)\big)$$
$$\leq \underbrace{2D_{KL}\big(\pi^*(a|s) \,\|\, \pi^0_A(a^0|s)\big)}_{\text{discrepancy of distributions}} + \underbrace{\frac{cK \log(2/\delta)}{N}}_{\text{error from Monte Carlo estimation}}.$$
$$(12)$$

*Here,* $\pi^*(a|s)$ *denotes the target Boltzmann distribution associated with the optimal dual variable* $\lambda^*$*, while* $\tilde{\pi}_A(a^0|s)$ *represents the approximated distribution generated from score function in* (11)*. See proof in Appendix C.3.*

Theorem 3.3 quantifies the statistical discrepancy between the diffusion sampled policy and the target Boltzmann distribution, providing a finite-sample approximation bound that complements the stability guarantees of ALGD. In practical sampling, the finite-step denoising procedure further introduces a discretization error. Following (Chen et al., 2022), this error can be expressed as $\epsilon(K, m, L_s)$, where $K$, $m$, and $L_s$ denote the number of denoising steps, the action dimension, and the Lipschitz constant of the score function, respectively. Hence, the total approximation error consists of the distribution discrepancy, the Monte Carlo estimation error, and the discretization error:

$$D_{\mathrm{KL}}\big(\pi^*(a|s) \,\|\, \tilde{\pi}^0_A(a^0|s)\big) \leq 2D_{\mathrm{KL}}\big(\pi^*(a|s) \,\|\, \pi^0_A(a^0|s)\big) +$$
$$\frac{cK \log(2/\delta)}{N} + \epsilon(K, m, L_s).$$

For our implementation, we set $K = 5$, and thus the discretization error is explicitly controlled by the smoothness of the score function and decreases polynomially as $K$ increases.

We remark that $D_{\mathrm{KL}}\big(\pi^*(a|s) \,\|\, \pi^0_A(a^0|s)\big)$ is bounded by $\mathcal{O}(\frac{\varepsilon^2}{\beta^2} + \frac{\varepsilon^2}{\beta^3} + \frac{\varepsilon^2}{\beta^4})$, with $|\mathcal{L}_A(s, a, \lambda) - \mathcal{L}(s, a, \lambda^*)| \leq \varepsilon$ for all $(s, a) \in \mathcal{S} \times \mathcal{A}$; see Corollary C.3 in Appendix C.3.

## 4. Numerical Experiments

To comprehensively evaluate performance, we conduct experiments on both the Safety-Gym benchmark (Ji et al., 2023) and the velocity-constrained MuJoCo benchmark (Tassa et al., 2018), which together represent a diverse set of safety-critical control tasks.

**Baselines.** Our baselines are selected to cover the paradigms in safe RL shown in Table 1. Specifically, we categorize

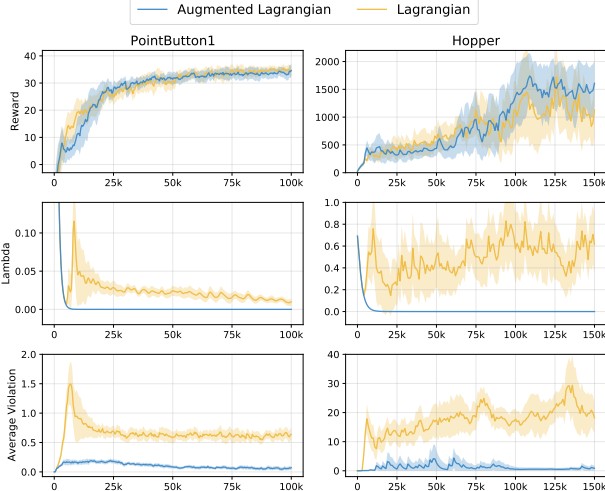

*Figure 2.* Comparative analysis of training stability between the standard Lagrangian (yellow) and the augmented Lagrangian (blue). (Top) Evaluation rewards; (Middle) Dual variable $\lambda$ updates; (Bottom) Average constraint violation (calculated as $\max(0, c(s) - h)$). The augmented formulation directly addresses the oscillation of dual variables (L2) and the instability of the induced Boltzmann distribution (L3) by regularizing the local curvature of the energy landscape. This results in more stable denoising dynamics and dual updating, allowing the policy to reach the safety threshold significantly faster and with zero dual variables (see full result in Figure 10 in Appendix E.6).

them into two classes. (1) *primal-dual methods*, which enforce safety constraints through Lagrangian relaxation and dual variable updates. This category includes both on-policy and off-policy algorithms, such as Conservative Augmented Lagrangian (CAL) (Wu et al., 2024), Soft Actor-Critic with Lagrangian (SAC + Lag) (Ji et al., 2023), Soft Actor-Critic with Augmented Lagrangian (SAC + AugLag), and Proximal Policy Optimization with Lagrangian relaxation (PPO + Lag) (Ji et al., 2023). (2) *Hard-constrained approaches*, which guarantee strict constraint satisfaction by explicitly characterizing the safe set via Hamilton-Jacobi (HJ) Rechability Analysis (Yu et al., 2022).

**Result analysis.** Our result analysis is structured around the core contributions of this work.

*Question 1: Can directly incorporating diffusion models into safe reinforcement learning resolve the key bottlenecks of this domain?* Our answer is **negative**. First, the energy landscape induced by safety constraints is highly irregular, which results in an unstable score field when diffusion models are applied directly. As illustrated in Figure 1 (left), the resulting energy surface is markedly non-smooth and exhibits sharp transitions near constraint boundaries. This severely degrades the reliability of the learned score function in high-cost regions. Because the denoising dynamics of diffusion models are governed by energy gradients, the

irregular energy landscape directly translates into unstable and unreliable policy sampling in online safe RL. Second, this challenge is further manifested in the training dynamics under standard Lagrangian optimization. As shown by the training curves in Figure 2 (yellow line), updates of the dual variables remain highly unstable, a direct consequence of the underlying irregular energy landscape. In particular, the policy frequently samples actions from regions that simultaneously yield high reward and high risk, leading to oscillations and repeated safety violations during training.

*Question 2: Why and how can our algorithm enable diffusion models to work in safe RL settings?* Our algorithm succeeds by explicitly addressing the interaction between diffusion dynamics and constrained optimization. Rather than relying on increased policy expressiveness, ALGD reshapes the energy landscape induced by safety constraints through an augmented Lagrangian formulation. This structural modification regularizes sharp transitions near constraint boundaries (see Figure 1, bottom), resulting in a smoother energy surface and a more stable score field for diffusion-based denoising. In addition, the augmented Lagrangian in ALGD introduces a quadratic penalty term, which biases policy sampling toward the safe region. By discouraging repeated visits to high-cost areas, this quadratic term reduces abrupt constraint violations during training, thereby yielding more stable primal-dual updates. As a result, both score learning and critic updating proceed in a smoother and more consistent manner, as reflected by the stable training dynamics shown in Figure 2 (blue line) and Figure 10 with full results.

*Question 3: Does our algorithm address the fundamental limitations of primal-dual methods in safe RL?* **Yes.** First, ALGD demonstrates markedly more stable training dynamics and consistently achieves competitive rewards with lower constraint costs across all evaluated baselines (see Figure 3). In contrast, existing off-policy primal-dual methods frequently suffer from oscillatory dual updates, which lead to unstable learning behavior and repeated sampling of policies that simultaneously yield high reward and high risk. For example, on the PointButton1&2 and HalfCheetah tasks, several primal-dual baselines attain relatively high rewards but incur substantial constraint violations, indicating non-convergent or poorly stabilized dual variables. Second, we observe that naively incorporating augmented Lagrangian techniques into the SAC framework does not yield consistent improvements. This is because the Gaussian policy parameterization used in SAC, even when combined with augmented Lagrangian penalties, does not fundamentally alter the underlying non-convex optimization geometry induced by safety constraints, the limitations (L1)-(L3) are still unsolved. In contrast, the diffusion-based policy representation in ALGD provides a substantially more expressive policy class when facing non-convex energy landscape.

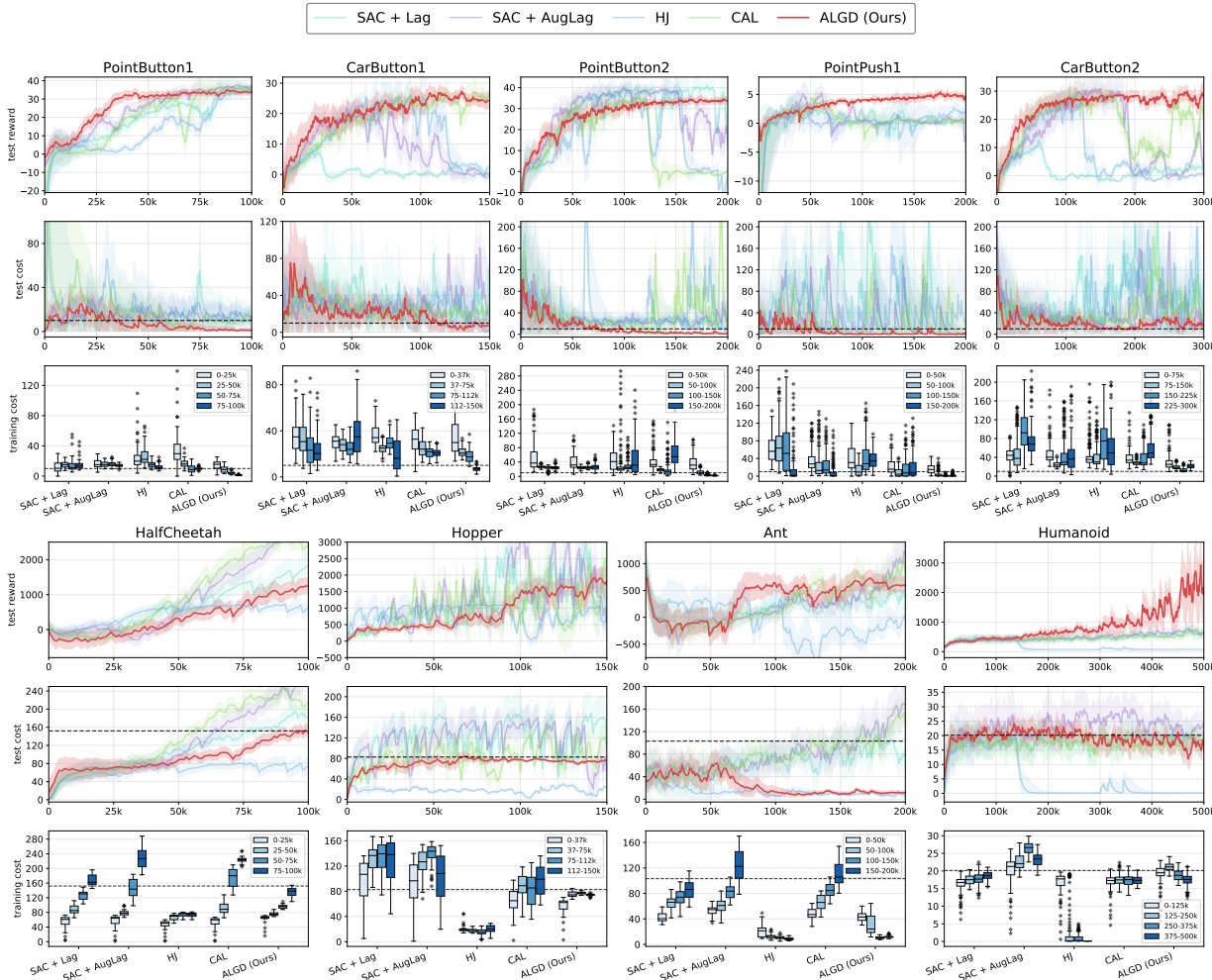

*Figure 3.* Comparison of performance across Safety-Gym and MuJoCo benchmarks. For each benchmark, the first row reports the evolution of test reward versus environment steps, the second row shows the corresponding test safety cost (with the dashed line indicating the cost budget), and the third row presents box plots of the training cost distribution over four equal step intervals. Overall, ALGD achieves competitive rewards while exhibiting improved stability and safety compared to baselines.

*Question 4: Does ALGD provide advantages over on-policy primal-dual and hard-constrained methods in safe RL?* **Yes.** Compared to on-policy primal-dual approaches such as PPO+Lag, ALGD exhibits higher sample efficiency due to its off-policy training paradigm (see Figure 6 in Appendix E.4). With significantly fewer environment interactions, ALGD consistently achieves higher rewards while maintaining lower constraint costs across all evaluated tasks. Moreover, unlike hard-constrained methods (i.e., HJ), ALGD does not rely on restricting the policy to an estimated safe set, which often leads to overly conservative behavior (low rewards and costs). As illustrated in Figure 3, this allows ALGD to explore a broader safe region and attain higher average returns without sacrificing safety. Importantly, ALGD satisfies the prescribed safety thresholds on all evaluated tasks, indicating empirical convergence to a saddle point of the constrained optimization problem.

**Ablation Study.** We conduct ablation studies on the number of Monte Carlo samples, critic ensemble size, and degree of convexification $\rho$ to validate our theory and robustness. We report partial results on the Monte Carlo sample size here. Full results are provided in Appendix E.5. These hyperparameters primarily serve to further improve performance; notably, our method remains effective and stable even without careful tuning of these choices.

*Monte Carlo Score Estimation.* Figure 4 shows that using more samples leads to more stable training dynamics and improved performance, characterized by lower variance and higher average returns. Moreover, since Monte Carlo sampling is performed in parallel, increasing the sample size incurs only a modest additional computational overhead, as shown in Table 2. This trend is consistent with our error analysis in Theorem 3.3, as increasing the number of

Monte Carlo samples improves score estimation, the error decreasing linearly as $\mathcal{O}(1/N)$.

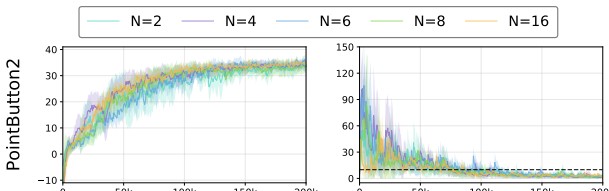

*Figure 4.* Ablation Studies of Monte Carlo sample size $N$. Due to the space limitation, see full results in Figure 7 in Appendix E.5.

*Table 2.* GPU time (ms) under sampling different Monte Carlo sample sizes $N$ (mean $\pm$ std).

| Task | $N=2$ | $N=4$ | $N=6$ | $N=8$ | $N=16$ |
|---|---|---|---|---|---|
| PointPush1 | 5.3±0.8 | 5.4±0.8 | 5.5±0.8 | 5.6±0.8 | 5.9±0.8 |
| PointButton2 | 4.6±1.0 | 4.7±1.0 | 4.8±0.8 | 4.9±1.1 | 5.0±1.1 |

## 5. Conclusion

We propose ALGD, a diffusion-based framework for online safe RL that uses an augmented Lagrangian to stabilize diffusion dynamics and primal-dual optimization. By convexifying the energy landscape without altering the optimal policy distribution, ALGD achieves competitive rewards with lower cost than prior methods.

While ALGD demonstrates strong empirical performance across a range of tasks, we currently do not provide formal sample complexity bounds or finite-sample convergence guarantees for the coupled dynamics arising from diffusion-based policy optimization and primal–dual updates. Analyzing such guarantees is challenging due to the interaction between stochastic diffusion processes, function approximation, and dual variable updates, and we leave a rigorous theoretical treatment to future work. Moreover, our experimental evaluation is restricted to simulated environments. Extending ALGD to real-world, safety-critical applications, where factors such as partial observability, model mismatch, and system latency play a significant role, remains an important direction for future research.

## Impact Statement

This work improves the safety and stability of diffusion-based reinforcement learning by enabling reliable policy learning under cost constraints in online and off-policy settings. By grounding diffusion policy sampling in augmented Lagrangian theory, the proposed method supports safer decision-making with expressive multimodal policies, which is beneficial for applications such as robotics and autonomous systems. Responsible deployment requires accurate specification of safety costs and appropriate system-level safeguards to ensure reliable behavior in real-world settings. We do not identify any specific ethical concerns beyond standard considerations.

## Acknowledgment

The corresponding author would like to thank the financial support provided by the Royal Academy of Engineering through the Industrial Fellowship scheme (IF-2425-19-AI165).

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

# Notation

| Notations | Meaning |
|:---:|:---:|
| $a$ | action |
| $a^{0\|\tau}$ | the posterior distribution $p(a^0\|a^\tau, s)$ |
| $c$ | safety cost |
| $d_0$ | initial distribution of state |
| $h$ | safety threshold |
| $r$ | reward function |
| $s$ | state |
| $t$ | time step |
| $\mathcal{A}$ | action set |
| $B^\tau$ | standard Brownian motions |
| $\mathcal{B}$ | replay buffer |
| $D_{\text{KL}}$ | KL divergence of two distributions |
| $\mathcal{L}$ | Lagrangian |
| $\mathcal{L}_A$ | augmented Lagrangian |
| $\mathcal{N}$ | Gaussian distribution |
| $Q$ | $Q-$function of reward |
| $Q_c$ | $Q-$function of cost |
| $\mathcal{S}$ | state set |
| $Z$ | partition function |
| $\pi$ | policy distribution induced by Lagrangian |
| $\pi_A$ | policy distribution induced by augmented Lagrangian |
| $\phi$ | score function |
| $\lambda$ | Lagrangian multiplier (a.k.a. dual variable) |
| $\tau$ | diffusion step |

# Appendix Overview

This appendix is organized into four main parts.

Part A presents a more comprehensive literature review on diffusion-based reinforcement learning, including additional related work omitted from the main text due to space constraints. Part **??** provides a discussion of the limitations of ALGD and highlights potential avenues for future work.

Part B provides a brief review of the classical *Lagrangian method* and its augmented variant used in constrained optimization, serving as the theoretical foundation of our formulation. Part C presents the detailed *theoretical analysis*, which consists of four interrelated results that build upon each other to establish the main claims of the paper.

Specifically,

- **Proposition 3.1** establishes the theoretical connection between diffusion-based score matching and the Lagrangian formulation of constrained reinforcement learning. It shows that, at optimality, the score field aligns with the gradient of the Lagrangian in the action space.

- **Theorem 3.2** extends this result by introducing the augmented Lagrangian and proving that the quadratic penalty term convexifies the local Lagrangian landscape while preserving the optimal policy and objective value of the original problem.

- **Theorem 3.3** quantifies the statistical discrepancy between the diffusion-generated policy and the target Boltzmann distribution, providing a finite-sample approximation bound that complements the structural stability guarantees derived from Theorem 3.2.

- **Corollary C.3** closes the analysis by showing that, as the augmented Lagrangian $\mathcal{L}_A$ converges to the true Lagrangian $\mathcal{L}$, the induced score mismatch and the resulting distributional gap vanish, thereby ensuring asymptotic consistency with the optimal constrained policy.

Together, these results form a coherent theoretical framework: the Proposition establishes the core connection between score matching and constrained optimization; the first Theorem provides structural and stability guarantees via the augmented formulation; the Error Analysis characterizes the finite-sample approximation error induced by diffusion-based sampling; and the final Theorem ensures that this residual gap shrinks to zero as the augmented Lagrangian recovers the true Lagrangian.

Regarding supplementary material for implementation and more experiments, Appendices D and E provide comprehensive details. Specifically, Appendix D elaborates on the implementation specifics of our proposed framework. In Appendix E, we extend our empirical evaluation with extensive comparisons against baseline on-policy algorithms. Furthermore, we conduct systematic ablation studies on the number of Monte Carlo samples, the critic ensemble size, and the degree of convexification $\rho$, thereby validating the robustness of our method and its alignment with the theoretical analysis. Moreover, training details are listed in Appendix E.2.

# A. More Literature Review

**Literature Review of Diffusion-based RL.** Recent advances in diffusion models (Song & Ermon, 2019; Ho et al., 2020; Song et al., 2020) have revealed fundamental connections between diffusion-based generative modeling and diffusion policies (Janner et al., 2022; Wang et al., 2022; Ren et al., 2024; Chi et al., 2025). However, these approaches are largely restricted to the offline setting. More recently, researchers have begun to explore diffusion policies in online RL settings, enabling continual interaction and policy improvement. One of the pioneering online diffusion-based RL methods is $Q$-score matching (Psenka et al., 2023), which uncovers an intrinsic connection between the score function in Langevin dynamics and the action-value gradient $\nabla_a Q(s, a)$. Following these this fundamental ideas, more research works explore the connections between reverse-time diffusion process and the structure $\nabla_a Q(s, a)$ (Ma et al., 2025; Dong et al., 2025). Unlike the previous idea, DACER (Wang et al., 2024) treats the reverse diffusion process as a direct policy function and employs a Gaussian-mixture entropy regulator to adaptively balance exploration and exploitation.

Despite recent progress, most existing diffusion-based approaches remain confined to the offline reinforcement learning setting. Recent studies have further explored safe offline policy generation using diffusion models (Xiao et al., 2023; Zheng et al., 2024; Cheng et al., 2025), primarily focusing on learning policies from static datasets and enforcing safety constraints during offline optimization. For example, (Xiao et al., 2023) incorporates control barrier functions to constrain diffusion-sampled trajectories, while (Zhang et al., 2025a) leverages optimization-based techniques and conditional diffusion mechanisms to generate constraint-satisfying policies. Beyond policy generation from fixed data distributions, Lyapunov-guided safe diffusion (Cheng et al., 2025) regularizes diffusion-sampled trajectories using Lyapunov functions; however, this approach assumes access to known and differentiable system dynamics. Overall, safe diffusion-based RL in the online setting, where policies are continuously updated through interaction with the environment under safety constraints, remains largely underexplored.

# B. Review Limitations of Original Lagrangian Method

Consider the optimization problem with inequality constraints:

$$\min_{x} \quad f(x)$$
$$\text{s.t.} \quad g_i(x) \leq 0, \quad i = 1, \ldots, m. \tag{13}$$

The *Lagrangian* is defined as (Boyd & Vandenberghe, 2004)

$$\mathcal{L}(x, \lambda) = f(x) + \sum_{i=1}^{m} \lambda_i g_i(x), \quad \lambda_i \geq 0. \tag{14}$$

The corresponding *dual problem* is written as

$$\max_{\lambda \succeq 0} \min_{x} \mathcal{L}(x, \lambda), \tag{15}$$

where the inner minimization over $x$ defines the dual function

$$q(\lambda) = \min_{x} \mathcal{L}(x, \lambda).$$

**Iterative Dual Updates.** A typical primal–dual method alternates between minimizing over $x$ and ascending over $\lambda$:

$$x^{(k+1)} = \arg\min_{x} \mathcal{L}(x, \lambda^{(k)}), \tag{16}$$

$$\lambda_i^{(k+1)} = \left[ \lambda_i^{(k)} + \eta_k g_i(x^{(k+1)}) \right]_+, \quad i = 1, \ldots, m, \tag{17}$$

where $\eta_k > 0$ is the step size and $[\,\cdot\,]_+ = \max(\,\cdot\,, 0)$ projects onto the nonnegative orthant.

This scheme first finds the primal variable minimizing the current Lagrangian, then updates the multipliers to penalize constraint violations, gradually approaching the saddle point of $\mathcal{L}(x, \lambda)$.

**Why Directly Using the Lagrangian in Diffusion Models Is Problematic.** The limitation of the standard Lagrangian formulation originates from its gradient structure. Given

$$\mathcal{L}(x, \lambda) = f(x) + \lambda g(x),$$

the primal update depends on the gradient

$$\nabla_x \mathcal{L}(x, \lambda) = \nabla_x f(x) + \lambda \nabla_x g(x).$$

When $f(x)$ and $g(x)$ are complex and represented by neural networks, this gradient field can be highly irregular and noisy, leading to instability in the minimization over $x$. Meanwhile, the dual update

$$\lambda^{(k+1)} = [\lambda^{(k)} + \eta_k g(x^{(k+1)})]_+$$

relies on possibly inaccurate evaluations of $g(x)$, which often cause oscillations or drift in $\lambda$. Because the penalty term $\lambda g(x)$ is purely linear, it provides no curvature regularization and fails to stabilize $\nabla_x \mathcal{L}(x, \lambda)$, resulting in fluctuating gradients and unreliable convergence in practice.

## C. Theoretical Analysis

Following the order of the main text, we provide the detailed proofs for all theoretical results presented in the paper. Each subsection corresponds to a specific result, including its assumptions, intermediate lemmas, and complete derivations. Specifically, we begin with the proof of Proposition 3.1, which establishes the optimality condition of the score function under the Lagrangian-guided diffusion formulation. We then present the proof of Theorem 3.2, demonstrating the convexification and invariance properties of the augmented Lagrangian. Finally, we provide the proof of Theorem 3.3, which bounds the approximation discrepancy between the diffusion-generated policy and the target Boltzmann distribution. Together, these proofs form a coherent theoretical foundation supporting the proposed Augmented Lagrangian-Guided Diffusion framework.

### C.1. Analysis of Proposition 3.1

To derive the variational form of the policy in (4), consider maximizing the entropy-regularized Lagrangian objective at each state $s$:

$$\max_{\pi(\cdot|s)} \mathbb{E}_{a\sim\pi(\cdot|s)}\big[Q^\pi(s,a) - \lambda(Q_c^\pi(s,a) - h) - \beta\log\pi(a|s)\big].$$

Defining Lagrangian $\mathcal{L}(s,a;\lambda) = -Q^\pi(s,a) + \lambda(Q_c^\pi(s,a) - h)$, the functional objective becomes

$$\mathcal{J}[\pi] = \int \pi(a|s)(\mathcal{L}(s,a;\lambda) + \beta\log\pi(a|s))\,da - \eta(\int \pi(a|s)\,da - 1).$$

Introducing a normalization constraint $\int \pi(a|s)\,da = 1$ with multiplier $\eta$ and setting the functional derivative to zero gives

$$\frac{\delta\mathcal{J}}{\delta\pi(a|s)} = \mathcal{L}(s,a;\lambda) + \beta(1 + \log\pi(a|s)) - \eta = 0.$$

Then, we have

$$\log\pi^*(a|s) \propto -\frac{\mathcal{L}(s,a,\lambda) + \eta}{\beta},$$

which yields the Boltzmann-form solution

$$\pi^*(a|s) = \frac{\exp\left(-\frac{\mathcal{L}(s,a,\lambda)}{\beta}\right)}{Z(s)}, \quad Z(s) = \int_a \exp\left(-\frac{\mathcal{L}(s,a,\lambda)}{\beta}\right)da, \tag{18}$$

indicating that the optimal policy follows a Boltzmann distribution, where fluctuations in the Lagrange multiplier $\lambda$ directly reshape the policy landscape.

In the following proposition, we present a method for estimating the exact score function for Lagrangian-guided diffusion under the VE SDE framework. We emphasize that the score function obtained via $Q$-score matching is generally *misspecified*: it computes the score as $\nabla_a Q(s, a^\tau)$ (Psenka et al., 2023), which is only valid at the final denoising step, i.e., when $\tau = 0$.

*Proof of Proposition 3.1.* According to the definition of the VE SDE (Chen et al., 2025), the intermediate distribution $\pi^\tau(a^\tau|s)$ is generated as

$$\pi^\tau(a^\tau|s) = \int \pi^0(a^0|s)\mathcal{N}\big(a^\tau; a^0, \sigma^2(\tau)I\big)\,da^0 = \big(\pi^0(a^0|s) * \mathcal{N}(0, \sigma^2(\tau)I)\big)(a^\tau),$$

which follows directly from the forward diffusion as a Gaussian smoothing of $\pi^0(a|s)$.

Differentiating under the integral sign yields the score of the intermediate distribution:

$$\nabla_a \log \pi^\tau(a^\tau|s) = \frac{\left(\nabla_a \pi^0(a^0|s) * \mathcal{N}(0, \sigma^2(\tau)I)\right)(a^\tau)}{\left(\pi^0(a^0|s) * \mathcal{N}(0, \sigma^2(\tau)I)\right)(a^\tau)}$$

$$= \frac{\mathbb{E}_{a^{0|\tau} \sim \mathcal{N}(a^\tau, \sigma^2(\tau)I)}\left[\nabla_a \pi(a^{0|\tau}|s)\right]}{\mathbb{E}_{a^{0|\tau} \sim \mathcal{N}(a^\tau, \sigma^2(\tau)I)}\left[\pi(a^{0|\tau}|s)\right]}. \tag{19}$$

The equality from the first line to the second line follows from the fact that differentiation (gradient) commutes with convolution (Brézis, 2011). Substituting the Boltzmann form $\pi(a^0|s) \propto \exp\left(-\frac{\mathcal{L}(s,a^0,\lambda)}{\beta}\right)$ and $\nabla_a \log \pi(a^0|s) = -\frac{\nabla_a \mathcal{L}(s,a^0,\lambda)}{\beta}$ gives

$$\frac{\mathbb{E}_{a^{0|\tau} \sim \mathcal{N}(a^\tau, \sigma^2(\tau)I)}\left[\nabla_a \pi(a^{0|\tau}|s)\right]}{\mathbb{E}_{a^{0|\tau} \sim \mathcal{N}(a^\tau, \sigma^2(\tau)I)}\left[\pi(a^{0|\tau}|s)\right]}$$

$$= \frac{\mathbb{E}_{a^{0|\tau} \sim \mathcal{N}(a^\tau, \sigma^2(\tau)I)}\left[-\frac{\exp\left(-\frac{\mathcal{L}(s,a^{0|\tau},\lambda)}{\beta}\right)}{\cancel{Z(s)}} \cdot \frac{\nabla_a \mathcal{L}(s,a^0,\lambda)}{\beta}\right]}{\mathbb{E}_{a^{0|\tau} \sim \mathcal{N}(a^\tau, \sigma^2(\tau)I)}\left[\frac{\exp\left(-\frac{\mathcal{L}(s,a^{0|\tau},\lambda)}{\beta}\right)}{\cancel{Z(s)}}\right]} \tag{20}$$

$$= \mathbb{E}_{a^{0|\tau} \sim \mathcal{N}(a^\tau, \sigma^2(\tau)I)}\left[-\frac{\exp\left(-\frac{\mathcal{L}(s,a^{0|\tau},\lambda)}{\beta}\right)}{\mathbb{E}_{a^{0|\tau} \sim \mathcal{N}(a^\tau, \sigma^2(\tau)I)}\left[\exp\left(-\frac{\mathcal{L}(s,a^{0|\tau},\lambda)}{\beta}\right)\right]} \frac{\nabla_a \mathcal{L}(s,a^{0|\tau},\lambda)}{\beta}\right].$$

**Connecting Score Matching with Policy Gradient.** In the Lagrangian formulation (4) of Safe RL, the optimal policy can be interpreted as a Boltzmann distribution defined by the energy function $\mathcal{L}(s, a, \lambda)$, i.e., $\pi(a|s) \propto \exp\left(-\frac{\mathcal{L}(s,a,\lambda)}{\beta}\right)$. Consequently, learning the optimal policy is equivalent to modeling the probability density induced by this Boltzmann distribution. From the diffusion perspective, the policy generation process

$$\underbrace{\pi^K(a^K|s)}_{\text{Gaussian prior}} \xrightarrow{\pi_\theta^K(a^{K-1}|a^K,s)} \pi^{K-1}(a^{K-1}|s) \xrightarrow{\pi_\theta^{K-1}(a^{K-2}|a^{K-1},s)} \cdots \xrightarrow{\pi_\theta^1(a^0|a^1,s)} \underbrace{\pi^0(a^0|s)}_{\text{target distribution}}$$

can be viewed as a reverse diffusion process that progressively transforms a Gaussian prior into the target Boltzmann policy. Each incremental refinement step, represented by the conditional transition $\pi_\theta^{\tau-1}(a^{\tau-1}|a^\tau, s)$, is guided by the score function $\nabla_a \log \pi^\tau(a^\tau|s)$, which governs the denoising dynamics. Since diffusion models are formulated via the reverse-time SDE in (2), optimizing the policy gradient term $\nabla_\theta \log \pi_\theta(a^{\tau-1}|a^\tau, s)$ at each refinement step can be interpreted as aligning the learned score field $\phi_\theta(s, a^\tau, \tau)$ with the intermediate score $\nabla_a \log \pi^\tau(a^\tau|s)$. In this sense, policy optimization approximates score matching within the reverse diffusion framework, linking Safe RL and diffusion modeling through a unified energy-based objective.

However, obtaining the exact form of the score function $\nabla_a \log \pi^\tau(a^\tau|s)$ is generally intractable, as it requires access to the intermediate marginal distribution $\pi^\tau(a^\tau|s)$ along the diffusion trajectory. To build a more practical and theoretically grounded connection between the score field and the Lagrangian energy $\mathcal{L}(s, a, \lambda)$, we derive in (20) an analytical form that explicitly links the two, revealing how the energy structure of Safe RL shapes the dynamics of the reverse-time SDE in diffusion-based policy optimization.

Therefore, the policy optimization in (6) achieves its optimal value when the score field satisfies

$$\phi_*(s, a^\tau, \tau) = \mathbb{E}_{a^{0|\tau} \sim \mathcal{N}(a^\tau, \sigma^2(\tau)I)}\left[-\frac{\exp\left(-\frac{\mathcal{L}(s,a^{0|\tau},\lambda)}{\beta}\right)}{\mathbb{E}_{a^{0|\tau} \sim \mathcal{N}(a^\tau, \sigma^2(\tau)I)}\left[\exp\left(-\frac{\mathcal{L}(s,a^{0|\tau},\lambda)}{\beta}\right)\right]} \frac{\nabla_a \mathcal{L}(s,a^{0|\tau},\lambda)}{\beta}\right].$$

This expression characterizes the *energy-guided score field*, where the gradient of the Lagrangian directly modulates the reverse diffusion dynamics that generate the policy.

## C.2. Proof of Theorem 3.2

**Proof.** We prove the claim in two parts. Part (a) shows that the quadratic penalty term in the augmented Lagrangian locally increases curvature in the action space, leading to improved conditioning of the energy landscape. Part (b) shows that, at optimality, the augmented Lagrangian preserves both the optimal objective value and the optimal policy of the original constrained problem.

**(a) Local Landscape Convexification.**

Recall the augmented Lagrangian

$$\mathcal{L}_A(s, a, \lambda) = -Q^\pi(s, a) + \frac{1}{2\rho}\Big([\lambda + \rho(Q_c^\pi(s, a) - h)]_+^2 - \lambda^2\Big),$$

where $\rho > 0$. We analyze the local curvature of $\mathcal{L}_A$ with respect to the action variable $a$.

We first consider the region where the constraint is locally active, i.e.,

$$\lambda + \rho\big(Q_c^\pi(s, a) - h\big) > 0.$$

In this case, the hinge operator reduces to the identity, and the augmented Lagrangian admits the equivalent form

$$\mathcal{L}_A(s, a, \lambda) = -Q^\pi(s, a) + \lambda\big(Q_c^\pi(s, a) - h\big) + \frac{\rho}{2}\big(Q_c^\pi(s, a) - h\big)^2 = \mathcal{L}(s, a, \lambda) + \frac{\rho}{2}\big(Q_c^\pi(s, a) - h\big)^2,$$

where $\mathcal{L}(s, a, \lambda) = -Q^\pi(s, a) + \lambda(Q_c^\pi(s, a) - h)$ denotes the standard Lagrangian.

Taking the Hessian with respect to $a$ yields

$$\nabla_a^2 \mathcal{L}_A(s, a, \lambda) = \nabla_a^2 \mathcal{L}(s, a, \lambda) + \rho\, \nabla_a Q_c^\pi(s, a)\nabla_a Q_c^\pi(s, a)^\top + \rho\big(Q_c^\pi(s, a) - h\big)\nabla_a^2 Q_c^\pi(s, a).$$

By assumption, $\nabla_a^2 Q_c^\pi$ is with smooth structure. Therefore, the last term is of order $\mathcal{O}(|Q_c^\pi(s, a) - h|)$. Consequently, in a neighborhood of the constraint boundary where $|Q_c^\pi(s, a) - h|$ is sufficiently small, the dominant curvature contribution introduced by the augmented term is

$$\rho\, \nabla_a Q_c^\pi(s, a)\nabla_a Q_c^\pi(s, a)^\top,$$

which is positive semidefinite. Hence, in this region,

$$\nabla_a^2 \mathcal{L}_A(s, a, \lambda) = \nabla_a^2 \mathcal{L}(s, a, \lambda) + \rho\, \nabla_a Q_c^\pi(s, a)\nabla_a Q_c^\pi(s, a)^\top + \mathcal{O}(|Q_c^\pi(s, a) - h|),$$

showing that the quadratic penalty strictly increases the local curvature of the energy landscape along the constraint-gradient directions. This curvature enhancement improves local conditioning and results in smoother gradients, which stabilizes denoising dynamics and score-based optimization.

When the constraint is inactive, i.e., $Q_c^\pi(s, a) \leq h - \frac{\lambda}{\rho}$, the hinge term vanishes and the augmented Lagrangian the quadratic penalty term vanishes since $[\lambda + \rho\big(Q_c^\pi(s, a) - h\big) > 0]_+ = 0$. In this case, no additional curvature is introduced and the local geometry remains unchanged. Thus, the convexifying effect of the augmented Lagrangian is both local and conditional.

**Discussion (When does local convexification occur?).** The above analysis shows that the convexifying effect of the augmented Lagrangian is *conditional* and *local*. Specifically, local convexification occurs only when the constraint is

active, i.e., $\lambda + \rho(Q_c^\pi(s,a) - h) > 0$, and in a neighborhood of the constraint boundary where $|Q_c^\pi(s,a) - h|$ is not large. In this regime, the quadratic penalty introduces a positive semidefinite curvature correction $\rho \nabla_a Q_c^\pi \nabla_a Q_c^{\pi\top}$, which locally reshapes the energy landscape and biases diffusion trajectories toward the feasible action set.

In contrast, when the constraint is inactive ($Q_c^\pi(s,a) \le h - \frac{\lambda}{\rho}$), the quadratic penalty vanishes and $\mathcal{L}_A$ coincides with the original Lagrangian $\mathcal{L}$. Therefore, no additional curvature is introduced and the local geometry remains unchanged. This selective convexification ensures that curvature enhancement is applied precisely where it is needed to prevent sampling unsafe actions, without over-regularizing the interior of the feasible region.

**(b) Invariance of the Optimal Policy and Objective.**

Since the original constrained problem is feasible and admits a primal-dual optimal solution satisfying the KKT conditions (e.g., under Slater's condition), we proceed to show that the augmented Lagrangian preserves both the optimal objective value and the optimal policy of the original constrained problem.

Let $(\pi^*, \lambda^*)$ denote a primal-dual solution satisfying the Karush–Kuhn–Tucker (KKT) conditions of the original problem (Nocedal & Wright, 2006). In particular, these conditions imply:

- *Primal feasibility:* $Q_c^{\pi^*}(s,a) \le h$ for $a \sim \pi^*$;

- *Dual feasibility:* $\lambda^* \ge 0$;

- *Complementary slackness:* $\lambda^*(Q_c^{\pi^*}(s,a) - h) = 0$ for $a \sim \pi^*$.

From complementary slackness, it follows that

$$[\lambda^* + \rho(Q_c^{\pi^*}(s,a) - h)]_+^2 - (\lambda^*)^2 = 0 \quad \text{for } a \sim \pi^*.$$

The augmented Lagrangian therefore yields

$$\mathcal{L}_A(s,a,\lambda^*) = \mathcal{L}(s,a,\lambda^*) \quad \forall a \in \text{supp}(\pi^*(\cdot|s)), \text{ with } Q_c^{\pi^*}(s,a) \le h.$$

As a consequence, the augmented objective preserves the optimal value:

$$\min_\pi \mathbb{E}_{s,a\sim\pi}\big[\mathcal{L}_A(s,a,\lambda^*)\big] = \min_\pi \mathbb{E}_{s,a\sim\pi}\big[\mathcal{L}(s,a,\lambda^*)\big].$$

Moreover, since the Boltzmann distribution induced by the energy function depends only on the value of the Lagrangian, the optimal policy distribution remains unchanged:

$$\pi_A^*(a|s) \propto \exp\Big(-\frac{\mathcal{L}_A(s,a,\lambda^*)}{\beta}\Big) = \exp\Big(-\frac{\mathcal{L}(s,a,\lambda^*)}{\beta}\Big) = \pi^*(a|s).$$

This completes the proof. The augmented Lagrangian reshapes the local energy landscape away from optimality to improve conditioning and sampling stability, while preserving both the optimal objective value and the optimal policy of the original constrained problem.

## C.3. Proof of Theorem 3.3 and Corollary C.3

**Lemma C.1** (Monte Carlo Score Estimation Error). *Assume that $\exp\left(-\frac{\mathcal{L}_A(s,a,\lambda)}{\beta}\right)$ and $\nabla_a \exp\left(-\frac{\mathcal{L}_A(s,a,\lambda)}{\beta}\right)$ are sub-Gaussian random variables over the Gaussian kernel $a^{0,(i)} \sim \mathcal{N}(a^\tau, \sigma^2(\tau)I)$. Let the Monte Carlo estimator of the score be*

$$\tilde{\phi}_A(s, a^\tau, \tau) = \sum_{i=1}^{N} -\frac{w_i}{\beta} \nabla_a \mathcal{L}_A(s, a^{0,(i)}, \lambda), \quad w_i = \frac{\exp\left[-\frac{1}{\beta}\mathcal{L}_A(s, a^{0,(i)}, \lambda)\right]}{\sum_{j=1}^{N} \exp\left[-\frac{1}{\beta}\mathcal{L}_A(s, a^{0,(j)}, \lambda)\right]}.$$

*Then there exists a constant $c(s, a^\tau)$ such that, with probability at least $1 - \delta$,*

$$\|\tilde{\phi}_A(s, a^\tau, \tau) - \phi_A(s, a^\tau, \tau)\| \leq \frac{c(s, a^\tau)\sqrt{\log(2/\delta)}}{\sqrt{N}}, \tag{21}$$

*where $\phi_A(s, a^\tau, \tau)$ denotes the true score function induced by augmented Lagrangian.*

*Proof.* Our target is to estimate $\nabla_a \log \pi_A^\tau(a^\tau|s)$ and $\mathbb{E}_{a^{0|\tau} \sim \mathcal{N}(a^\tau, \sigma^2(\tau)I)}\left[\exp(-\frac{\mathcal{L}_A(s, a^{0|\tau}, \lambda)}{\beta})\right] \propto \pi_A^\tau(a^\tau|s)$. When assume that the random variables $\exp\left(-\frac{\mathcal{L}_A(s, a^{0,(i)}, \lambda)}{\beta}\right)$ and $\nabla_a \exp\left(-\frac{\mathcal{L}_A(s, a^{0,(i)}, \lambda)}{\beta}\right)$ with $a^{0,(i)} \sim \mathcal{N}(a^\tau, \sigma^2(\tau)I)$ are sub-Gaussian [1], then by Hoeffding's inequality on sub-Gaussian random variables (Rigollet & Hütter, 2023), we have that there exists a constant $C > 0$ such that for any $\delta > 0$ with probability $1 - \delta$

$$\left|\frac{1}{N}\sum_{i=1}^{N} \exp\left(-\frac{\mathcal{L}_A(s, a^{0,(i)}, \lambda)}{\beta}\right) - \exp\left(-\frac{\mathcal{L}_A(s, a^\tau, \lambda)}{\beta}\right)\right| \leq C\frac{\sqrt{\log(2/\delta)}}{\sqrt{N}} \tag{22}$$

and

$$\left\|\frac{1}{N}\sum_{i=1}^{N} \nabla_a \exp\left(-\frac{\mathcal{L}_A(s, a^{0,(i)}, \lambda)}{\beta}\right) - \nabla_a \exp\left(-\frac{\mathcal{L}_A(s, a^\tau, \lambda)}{\beta}\right)\right\| \leq C\frac{\sqrt{\log(2/\delta)}}{\sqrt{N}}. \tag{23}$$

Substituting the true score function, we have the following result with probability $1 - \delta$

$$\begin{aligned}
&\|\tilde{\phi}(s, a^\tau, \tau) - \phi_A(s, a^\tau, \tau)\| \\
&= \left\|\frac{\frac{1}{N}\sum_{i=1}^{N}\nabla_a\exp\left(-\frac{\mathcal{L}_A(s,a^{0,(i)},\lambda)}{\beta}\right)}{\frac{1}{N}\sum_{i=1}^{N}\exp\left(-\frac{\mathcal{L}_A(s,a^{0,(i)},\lambda)}{\beta}\right)} - \frac{\nabla_a\exp\left(-\frac{\mathcal{L}_A(s,a^\tau,\lambda)}{\beta}\right)}{\exp\left(-\frac{\mathcal{L}_A(s,a^\tau,\lambda)}{\beta}\right)}\right\| \\
&= \left\|\frac{\frac{1}{N}\sum_{i=1}^{N}\nabla_a\exp\left(-\frac{\mathcal{L}_A(s,a^{0,(i)},\lambda)}{\beta}\right)\exp\left(-\frac{\mathcal{L}_A(s,a^\tau,\lambda)}{\beta}\right) - \frac{1}{N}\sum_{i=1}^{N}\exp\left(-\frac{\mathcal{L}_A(s,a^{0,(i)},\lambda)}{\beta}\right)\nabla_a\exp\left(-\frac{\mathcal{L}_A(s,a^\tau,\lambda)}{\beta}\right)}{\exp\left(-\frac{\mathcal{L}_A(s,a^\tau,\lambda)}{\beta}\right)\frac{1}{N}\sum_{i=1}^{N}\exp\left(-\frac{\mathcal{L}_A(s,a^{0,(i)},\lambda)}{\beta}\right)}\right\| \\
&\leq \left\|\frac{\frac{1}{N}\sum_{i=1}^{N}\nabla_a\exp\left(-\frac{\mathcal{L}_A(s,a^{0,(i)},\lambda)}{\beta}\right) - \nabla_a\exp\left(-\frac{\mathcal{L}_A(s,a^\tau,\lambda)}{\beta}\right)}{\sum_{i=1}^{N}\exp\left(-\frac{\mathcal{L}_A(s,a^{0,(i)},\lambda)}{\beta}\right)}\right\| \\
&\quad + \left\|\nabla_a\exp\left(-\frac{\mathcal{L}_A(s,a^\tau,\lambda)}{\beta}\right)\cdot\frac{\exp\left(-\frac{\mathcal{L}_A(s,a^\tau,\lambda)}{\beta}\right) - \frac{1}{N}\sum_{i=1}^{N}\exp\left(-\frac{\mathcal{L}_A(s,a^{0,(i)},\lambda)}{\beta}\right)}{\exp\left(-\frac{\mathcal{L}_A(s,a^\tau,\lambda)}{\beta}\right)\sum_{i=1}^{N}\frac{1}{N}\exp\left(-\frac{\mathcal{L}_A(s,a^{0,(i)},\lambda)}{\beta}\right)}\right\|.
\end{aligned} \tag{24}$$

Under regularity assumption $\exp(-\frac{\mathcal{L}_A(s,a,\lambda)}{\beta}) \geq m > 0$ for all $(s, a) \in \mathcal{S} \times \mathcal{A}$ and $\sup_{s,a} \|\nabla_a \exp(-\frac{\mathcal{L}_A(s,a,\lambda)}{\beta})\| \leq M < \infty$, then combining to the inequality (24) and applying union bound to the two concentration inequalities, there exists a universal constant $c(s, a^\tau) \geq \frac{1}{m} + \frac{M}{m^2}$ such that we have with probability $1 - 2\delta$

$$\|\tilde{\phi}(s, a^\tau, \tau) - \phi_A(s, a^\tau, \tau)\| \leq \frac{c(s, a^\tau)\sqrt{\log(2/\delta)}}{\sqrt{N}}. \tag{25}$$

---

[1]Although Gaussian kernels with infinite support are used for score estimation, the effective domain of integration is restricted to the compact action space $\mathcal{A}$. As a result, the induced Boltzmann distribution is supported on a bounded domain and is therefore sub-Gaussian. The additional assumption that $\mathcal{L}_A(s, a)$ has bounded gradient on $\mathcal{A}$ ensures that the corresponding score function is also sub-Gaussian.

**Lemma C.2** (Pathwise KL Divergence under Drift Perturbation via Girsanov). *Let $x^\tau$ satisfy the Itô SDEs*

$$dx^\tau = b_1(x^\tau)\,d\tau + \sigma(x^\tau)\,dB^\tau,$$
$$dy^\tau = b_2(y^\tau)\,d\tau + \sigma(y^\tau)\,dB^\tau,$$

*where $B^\tau$ is a standard Brownian motion on $[0, T]$, and the diffusion matrix $\sigma(x)$ is uniformly non-degenerate and identical for both dynamics. Let $p_1$ and $p_2$ denote the path measures on $C([0, T]; \mathbb{R}^d)$ induced by these two SDEs. Assume that $b_1, b_2, \sigma$ satisfy the usual Lipschitz and linear growth conditions ensuring strong existence and uniqueness of solutions.*

*Then $p_2$ is absolutely continuous with respect to $p_1$, and the Kullback–Leibler divergence between the two path measures satisfies*

$$D_{KL}(p_2 \,\|\, p_1) = \frac{1}{2}\,\mathbb{E}_{p_2}\left[\int_0^T \left\|\sigma^{-1}(b_2(x^\tau) - b_1(x^\tau))\right\|^2 d\tau\right]. \tag{26}$$

*Proof.* By Girsanov's theorem, since the diffusion coefficients of the two SDEs coincide and $\sigma(x)$ is invertible, the path measure $p_2$ is absolutely continuous with respect to $p_1$. The Radon–Nikodym derivative of $p_2$ with respect to $p_1$ on the path space is given by

$$\frac{dp_2}{dp_1} = \exp\left(\int_0^T (\sigma^{-1}(b_2 - b_1))^\top dB^\tau - \frac{1}{2}\int_0^T \|\sigma^{-1}(b_2 - b_1)\|^2\,d\tau\right),$$

where all quantities are evaluated along the trajectory $x^\tau$ under $p_1$.

By definition of the Kullback–Leibler divergence,

$$D_{\mathrm{KL}}(p_2 \,\|\, p_1) = \mathbb{E}_{p_2}\left[\log\left(\frac{dp_2}{dp_1}\right)\right].$$

Under $p_2$, the stochastic integral $\int_0^T (\sigma^{-1}(b_2 - b_1))^\top dB^\tau$ has zero mean. Therefore,

$$D_{\mathrm{KL}}(p_2 \,\|\, p_1) = \frac{1}{2}\,\mathbb{E}_{p_2}\left[\int_0^T \|\sigma^{-1}(b_2(x^\tau) - b_1(x^\tau))\|^2\,d\tau\right],$$

which establishes the result.

---

*Proof.* Returning to our setting, we adopt a *same-path construction* for the reverse-time diffusion process. Specifically, we consider a single stochastic trajectory $a^\tau \in C([0, K]; \mathbb{R}^d)$ defined on a common filtered probability space and driven by the same Brownian motion $B^\tau$. Different policies are induced by evaluating different drift fields along this same path.

Concretely, the reverse-time dynamics take the unified form

$$da^\tau = -\frac{d\sigma^2(\tau)}{d\tau}\,\phi(s, a^\tau, \tau)\,d\tau + \sqrt{\frac{d\sigma^2(\tau)}{d\tau}}\,dB^\tau,$$

where the drift field $\phi$ is instantiated as one of $\tilde{\phi}_A$, $\hat{\phi}_A$, or $\phi_*$. The diffusion coefficient $\sqrt{\frac{d\sigma^2(\tau)}{d\tau}}$ is assumed to be uniformly non-degenerate and identical across all cases.

Here, $\tilde{\phi}_A(s, a^\tau, \tau)$ denotes a Monte Carlo score estimator constructed from the augmented Lagrangian $\mathcal{L}_A$, which induces the approximate policy $\tilde{\pi}_A^0$; $\hat{\phi}_A(s, a^\tau, \tau)$ denotes the exact score field induced by $\mathcal{L}_A$, which induces the intermediate policy $\pi_A^0$; and $\phi_*(s, a^\tau, \tau)$ corresponds to the optimal score associated with the true Lagrangian $\mathcal{L}(s, a, \lambda^*)$, inducing the target Boltzmann policy $\pi^*$ (or $\pi^0(a^0|s)$ as clean distribution).

By Lemma C.2 and the same-path construction above, the pathwise KL divergence between the distributions induced by

$\tilde{\phi}_A$ and $\phi_*$ admits the representation

$$D_{\mathrm{KL}}\big(\pi^0(a^0|s) \,\|\, \tilde{\pi}_A^0(a^0|s)\big) = \frac{1}{2}\, \mathbb{E}_{\pi^0(a^0|s)}\left[\int_0^K \left\|\left(\sqrt{\frac{d\sigma^2(\tau)}{d\tau}}\right)^{-1} \times \frac{d\sigma^2(\tau)}{d\tau}\big(\tilde{\phi}_A(s,a^\tau,\tau) - \phi_*(s,a^\tau,\tau)\big)\right\|^2 d\tau\right]$$

$$= \frac{1}{2}\, \mathbb{E}_{\pi^0(a^0|s)}\left[\int_0^K \left\|\sqrt{\frac{d\sigma^2(\tau)}{d\tau}}\big(\tilde{\phi}_A(s,a^\tau,\tau) - \phi_*(s,a^\tau,\tau)\big)\right\|^2 d\tau\right]$$

$$(27)$$

To isolate different sources of approximation error, we decompose the drift discrepancy as

$$\tilde{\phi}_A - \phi_* = (\tilde{\phi}_A - \hat{\phi}_A) + (\hat{\phi}_A - \phi_*),$$

and apply the inequality $\|x+y\|^2 \leq 2\|x\|^2 + 2\|y\|^2$, which is valid due to the quadratic form of the pathwise KL divergence. This yields

$$D_{\mathrm{KL}}\big(\pi^0 \,\|\, \tilde{\pi}_A^0\big) \leq \mathbb{E}_{\pi^0} \int_0^K \left\|\sqrt{\frac{d\sigma^2(\tau)}{d\tau}}\big(\tilde{\phi}_A - \hat{\phi}_A\big)\right\|^2 d\tau$$

$$+ \underbrace{\mathbb{E}_{\pi^0} \int_0^K \left\|\sqrt{\frac{d\sigma^2(\tau)}{d\tau}}\big(\hat{\phi}_A - \phi_*\big)\right\|^2 d\tau}_{=2D_{\mathrm{KL}}\big(\pi^0 \,\|\, \pi_A^0\big)}. \qquad (28)$$

The first term in (28) captures the error induced by Monte Carlo score estimation, and the second term is equal to $2D_{\mathrm{KL}}(\pi^0 \,\|\, \pi_A^0)$ according to (26). Given the sub-Gaussianity assumption, the high-probability bound can be integrated to yield the expected error bound. For $N$ samples, Lemma C.1 implies that, with probability at least $1 - 2\delta$,

$$\mathbb{E}_{\pi^0} \int_0^K \left\|\sqrt{\frac{d\sigma^2(\tau)}{d\tau}}\big(\tilde{\phi}_A - \hat{\phi}_A\big)\right\|^2 d\tau \leq \frac{c_1 c_2 K \log(1/\delta)}{N}, \qquad (29)$$

where we have nondegenerate noise

$$c_1 := \sup_{\tau \in [0,K],\, a \in \mathcal{A}} c(s,a^\tau), \qquad c_2 := \sup_{\tau \in [0,K]} \frac{d\sigma^2(\tau)}{d\tau}$$

Finally, as $\mathcal{L}_A(s,a,\lambda) \to \mathcal{L}(s,a,\lambda^*)$, the second term in (28) vanishes. Consequently, the discrepancy between the approximate diffusion policy $\tilde{\pi}_A^0$ and the optimal policy $\pi^*(a^0|s)$ is dominated by the Monte Carlo path integration error, which decays at the rate $\mathcal{O}\big(\frac{\log(1/\delta)}{N}\big)$. $\qquad \square$

To complete the analysis, we next relate the discrepancy from the target Boltzmann distribution to the approximation gap $\mathcal{L}_A(s,a,\lambda) - \mathcal{L}(s,a,\lambda^*)$, showing how deviations in the augmented Lagrangian translate into distributional mismatch.

**Corollary C.3** (Stability of score under Lagrangian approximation). *Let the target score function at time $\tau$ with optimal dual variable $\lambda^*$ be defined as*

$$\phi_*(s,a^\tau,\tau) = \mathbb{E}_{a^{0|\tau} \sim \mathcal{N}(a^\tau, \sigma^2(\tau)I)}\left[-\frac{\exp\left(-\frac{\mathcal{L}(s,a^{0|\tau},\lambda^*)}{\beta}\right)}{\mathbb{E}_{a^{0|\tau}}\left[\exp\left(-\frac{\mathcal{L}(s,a^{0|\tau},\lambda^*)}{\beta}\right)\right]} \frac{\nabla_a \mathcal{L}(s,a^{0|\tau},\lambda^*)}{\beta}\right]. \qquad (30)$$

*Let $\mathcal{L}_A(s,a,\lambda)$ be an approximate augmented Lagrangian. Assume that there exists $\varepsilon > 0$ such that*

$$\|\mathcal{L}_A(s,a,\lambda) - \mathcal{L}(s,a,\lambda^*)\|_\infty \leq \varepsilon, \qquad (31)$$

*and, in addition, that the approximation is stable at the gradient level:*

$$\|\nabla_a \mathcal{L}_A(s,a,\lambda) - \nabla_a \mathcal{L}(s,a,\lambda^*)\|_\infty \leq \varepsilon. \qquad (32)$$

*Assume further that the gradients are uniformly bounded,*

$$\|\nabla_a \mathcal{L}_A(s, a, \lambda)\|, \ \|\nabla_a \mathcal{L}(s, a, \lambda^*)\| \le G.$$

*Let $\hat{\phi}_A(s, a^\tau, \tau)$ denote the score induced by $\mathcal{L}_A$ via the same definition as (30). Then there exist constants $C_1, C_2 > 0$, independent of $\beta$ and $\varepsilon$, such that*

$$\left\| \phi_*(s, a^\tau, \tau) - \hat{\phi}_A(s, a^\tau, \tau) \right\|^2 \le C_1^2 \frac{\varepsilon^2}{\beta^2} + 2 C_1 C_2 \frac{\varepsilon^2}{\beta^3} + C_2^2 \frac{\varepsilon^2}{\beta^4}. \tag{33}$$

*Consequently, the discrepancy from the target Boltzmann distribution induced by the score mismatch vanishes as $\varepsilon \to 0$, with a rate explicitly controlled by the temperature parameter $\beta$.*

*Proof.* Recall that the target score function with the optimal dual variable $\lambda^*$ is defined as

$$\phi_*(s, a^\tau, \tau) = \mathbb{E}_{a^{0|\tau} \sim \mathcal{N}(a^\tau, \sigma^2(\tau)I)} \left[ - \frac{\exp\left( -\frac{\mathcal{L}(s, a^{0|\tau}, \lambda^*)}{\beta} \right)}{\mathbb{E}\left[ \exp\left( -\frac{\mathcal{L}(s, a^{0|\tau}, \lambda^*)}{\beta} \right) \right]} \frac{\nabla_a \mathcal{L}(s, a^{0|\tau}, \lambda^*)}{\beta} \right]. \tag{34}$$

Let $\hat{\phi}_A(s, a^\tau, \tau)$ denote the score induced by the approximate augmented Lagrangian $\mathcal{L}_A(s, a, \lambda)$ via the same definition.

**Step 1: Convergence of the augmented Lagrangian.** Since $\lambda^*$ is the optimal dual variable, by the KKT conditions the augmented Lagrangian coincides with the standard Lagrangian at optimality. Hence, as $\lambda \to \lambda^*$,

$$\mathcal{L}_A(s, a, \lambda) \to \mathcal{L}(s, a, \lambda^*). \tag{35}$$

By assumption, the approximation error satisfies

$$\left| \mathcal{L}_A(s, a, \lambda) - \mathcal{L}(s, a, \lambda^*) \right| \le \varepsilon, \quad \text{for all } (s, a) \in \mathcal{S} \times \mathcal{A}. \tag{36}$$

**Step 2: Unified score representation.** For any energy induced by Lagrangian $\mathcal{L}(s, a, \lambda^*)$, define the normalized probability density as

$$w_\mathcal{L}(a^{0|\tau}) = \frac{\exp\left( -\frac{\mathcal{L}(s, a^{0|\tau}, \lambda^*)}{\beta} \right)}{\mathbb{E}_{a^{0|\tau} \sim \mathcal{N}(a^\tau, \sigma^2(\tau)I)} \left[ \exp\left( -\frac{\mathcal{L}(s, a^{0|\tau}, \lambda^*)}{\beta} \right) \right]}. \tag{37}$$

Then the score can be written as

$$\phi_*(s, a^\tau, \tau) = \mathbb{E}_{a^{0|\tau}} \left[ - w_\mathcal{L}(a^{0|\tau}) \frac{\nabla_a \mathcal{L}(s, a^{0|\tau}, \lambda^*)}{\beta} \right]. \tag{38}$$

In particular,

$$\phi_* = \phi_{\mathcal{L}(\cdot, \lambda^*)}, \qquad \hat{\phi}_A = \phi_{\mathcal{L}_A(\cdot, \lambda)}. \tag{39}$$

**Step 3: Error decomposition.** We decompose the difference as

$$\phi_* - \hat{\phi}_A = \mathbb{E}_{a^{0|\tau}} \left[ \left( w_{\mathcal{L}_A} - w_\mathcal{L} \right) \frac{\nabla_a \mathcal{L}}{\beta} + w_{\mathcal{L}_A} \frac{\nabla_a \mathcal{L}_A - \nabla_a \mathcal{L}}{\beta} \right]. \tag{40}$$

**Step 4: Bounding the gradient discrepancy term.** By differentiability and assumption (32), there exists a constant $C > 0$ such that

$$\|\nabla_a \mathcal{L}(s, a, \lambda^*) - \nabla_a \mathcal{L}_A(s, a, \lambda)\| \le C \varepsilon. \tag{41}$$

Since $0 \le w_{\mathcal{L}_A} \le 1$, we obtain

$$(\text{II}) \le \mathbb{E}\left[ \frac{C \varepsilon}{\beta} \right] = C_1 \frac{\varepsilon}{\beta}, \tag{42}$$

where $C_1 > 0$ is a constant.

**Step 5: Bounding the probability density discrepancy.** From (36), we have

$$\exp\left(-\frac{\mathcal{L}_A}{\beta}\right) = \exp\left(-\frac{\mathcal{L}}{\beta}\right)\left(1 + \mathcal{O}\left(\frac{\varepsilon}{\beta}\right)\right). \tag{43}$$

By standard softmax stability arguments (Mitzenmacher & Upfal, 2017), this implies

$$\|w_{\mathcal{L}_A} - w_{\mathcal{L}}\| \le C\,\frac{\varepsilon}{\beta}. \tag{44}$$

Using the bounded gradient assumption $\|\nabla_a \mathcal{L}\| \le G$, we obtain

$$(\mathrm{I}) \le \mathbb{E}\left[\frac{G}{\beta} \cdot \frac{\varepsilon}{\beta}\right] = C_2 \frac{\varepsilon}{\beta^2}, \tag{45}$$

where $C_2 > 0$ is a constant.

**Step 6: Final bound.** Combining (42) and (45) yields

$$\left\|\phi_*(s, a^\tau, \tau) - \hat{\phi}_A(s, a^\tau, \tau)\right\|^2 \le C_1^2 \frac{\varepsilon^2}{\beta^2} + 2C_1 C_2 \frac{\varepsilon^2}{\beta^3} + C_2^2 \frac{\varepsilon^2}{\beta^4}. \tag{46}$$

This completes the proof.

*Remark* C.4. As established in the preceding analysis, the discrepancy from the target Boltzmann distribution, $D_{\mathrm{KL}}\big(\pi^*(a^0|s) \,\|\, \pi_A^0(a^0|s)\big)$, is directly controlled by the approximation gap between the augmented Lagrangian $\mathcal{L}_A(s, a, \lambda)$ and the true Lagrangian $\mathcal{L}(s, a, \lambda^*)$. In particular, as $\varepsilon \to 0$, this discrepancy converges to zero at the rate $\mathcal{O}\big(\frac{\varepsilon^2}{\beta^2} + \frac{\varepsilon^2}{\beta^3} + \frac{\varepsilon^2}{\beta^4}\big)$.

# D. Implementation and Pseudo Code

Algorithm 1 presents the augmented Lagrangian-guided diffusion (ALGD) algorithm. This framework can also be applied to the standard Lagrangian-guided diffusion (LGD) method; the only required modification is to replace the augmented Lagrangian term $\mathcal{L}_A$ with the standard Lagrangian $\mathcal{L}$ throughout Algorithm 1.

Algorithm 1 summarizes the augmented Lagrangian-guided diffusion (ALGD) framework. At each environment interaction step, actions are sampled via a reverse-time SDE conditioned on the current state. Starting from Gaussian noise, the action is iteratively denoised using the learned score network, yielding a final action $a_t^0$ that is executed in the environment.

During training, the Q-value critics and cost critics are updated using standard temporal-difference learning. The reward critics are trained with a clipped double-Q objective, while an ensemble of cost critics is used to estimate the expected constraint cost, improving robustness and reducing variance.

The score network is updated by matching its output to a Monte Carlo estimate of the true score induced by the augmented Lagrangian objective. Specifically, the augmented Lagrangian combines the reward maximization objective with a penalty term that enforces the cost constraint via a dynamically updated Lagrange multiplier. Gradients of this augmented objective with respect to the action are used to construct a target score, and the score network is trained by minimizing a mean-squared error loss. Finally, the Lagrange multiplier is updated using projected gradient ascent to ensure constraint satisfaction.

**Numerical Stability in Score Estimation.**   In our implementation, we address the potential numerical instability when computing the Boltzmann weights $w_i$ defined in (11). Due to the quadratic penalty term $\rho$ in the augmented Lagrangian $\mathcal{L}_A$, the energy values can reach a large value. To mitigate this, we employ the *Log-Sum-Exp* trick. Specifically, for a batch of $N$ Monte Carlo samples $\{a^{0|\tau,(i)}\}_{i=1}^N$, we define the intermediate energy terms as:

$$E_i = -\frac{\mathcal{L}_A(s, a^{0|\tau,(i)}, \lambda)}{\beta}. \tag{47}$$

The weights are then computed as:

$$w_i = \frac{\exp(E_i - \max_j E_j)}{\sum_{k=1}^N \exp(E_k - \max_j E_j)}. \tag{48}$$

This formulation ensures that the argument of the exponential function is at most $0$, thereby preventing overflow while preserving the relative proportions of the weights. This stabilized weighting mechanism is crucial for maintaining a reliable score field $\phi_A(s, a^\tau, \tau)$ during the early stages of training when the dual variable $\lambda$ or the penalty $\rho$ may induce sharp energy transitions.

**Amortized policy generation.**   Although the Monte Carlo estimation in the score update involves sampling multiple action trajectories, this computation is only performed during *training* of the score network. At interaction time, policy generation is fully amortized: actions are produced by a diffusion sampling pass using the learned score network, without requiring any Monte Carlo sampling. As a result, environment interaction incurs no additional computational overhead.

---

**Algorithm 1** Augmented Lagrangian-Guided Diffusion (ALGD)

---

**Require:** Replay buffer $\mathcal{B}$; double Q-networks $\{Q^{\phi_1}, Q^{\phi_2}\}$; ensemble cost critics $\{Q_c^{\psi_i}\}_{i=1}^M$; score network $\phi_\theta$; Lagrange multiplier $\lambda$; diffusion steps $K$; Monte Carlo sample number $N$; noise schedule $\sigma(\tau)$; temperature $\beta$

1: Initialize parameters $\phi_1, \phi_2, \{\psi_i\}_{i=1}^M, \theta$, and $\lambda \geq 0$
2: **for** each environment step **do**
3:      Observe state $s_t$
4:      Sample noisy action $a_t^K \sim \mathcal{N}(0, \sigma^2(K)I)$
5:      **for** $\tau = K$ **down to** 1 **do**
6:          Diffusion sampling via score network $\phi_\theta(s_t, a_t^\tau, \tau)$

$$a_t^{\tau-1} = a_t^\tau + \frac{d\sigma^2(\tau)}{d\tau} \phi_\theta(s_t, a_t^\tau, \tau) + \sqrt{\frac{d\sigma^2(\tau)}{d\tau}} \epsilon, \quad \epsilon \sim \mathcal{N}(0, I)$$

7:      **end for**
8:      Execute $a_t^0$, observe $(r_t, c_t, s_{t+1})$
9:      Store $(s_t, a_t^0, r_t, c_t, s_{t+1})$ in $\mathcal{B}$
10: **end for**
11: **for** each gradient update step **do**
12:      Sample batch $(s, a, r, c, s') \sim \mathcal{B}$
13:      Sample next action $a' \sim \pi_\theta(\cdot|s')$ via diffusion sampling
14:      Compute target Q-value

$$y_Q = r + \gamma \min_{j=1,2} Q^{\phi_j'}(s', a')$$

15:      Compute target cost value

$$y_{Q_c} = c + \gamma \frac{1}{M} \sum_{i=1}^M Q_c^{\psi_i'}(s', a')$$

16:      Update critics

$$\phi_j \leftarrow \phi_j - \eta_Q \nabla_{\phi_j} (Q^{\phi_j}(s, a) - y_Q)^2, \quad j = 1, 2$$

$$\psi_i \leftarrow \psi_i - \eta_Q \nabla_{\psi_i} (Q_c^{\psi_i}(s, a) - y_{Q_c})^2$$

17:      Compute ensemble cost critic

$$\bar{Q}_c(s, a) = \frac{1}{M} \sum_{i=1}^M Q_c^{\psi_i}(s, a)$$

18:      **Score network update**
19:      Monte Carlo target score estimation with the *Log-Sum-Exp* trick

$$\phi_A(s, a^\tau, \tau) \approx \sum_{i=1}^N -\frac{w_i}{\beta} \nabla_a \mathcal{L}_A(s, a^{0|\tau,(i)}, \lambda), \quad a^{0|\tau,(i)} \sim \mathcal{N}(a^\tau, \sigma^2(\tau))$$

20:      Minimize score loss

$$\mathcal{L}_{\text{score}}(\theta) = \mathbb{E}\left[ \|\phi_\theta(s, a^\tau, \tau) - \phi_A(s, a^\tau, \tau)\|^2 \right]$$

21:      Update Lagrange multiplier

$$\lambda \leftarrow [\lambda + \eta_\lambda(\bar{Q}_c(s, a) - h)]_+$$

22: **end for**

---

# E. More Experimental Results

## E.1. Experimental Settings

We evaluate our method on a suite of safe RL benchmarks that require agents to achieve task objectives while strictly respecting safety constraints. As shown in Figure 5, the environments span both structured manipulation/navigation scenarios and complex control tasks, providing a comprehensive testbed for safe RL algorithms. To accelerate training, we adopted the modifications made to the Safety-Gym task by (Liu et al., 2022a). Since all algorithms are evaluated on the same set of tasks, the comparisons remain valid.

**Safe-Gym Environments.** The Safe-Gym tasks, including *Point Button*, *Car Button*, and *Point Push*, impose explicit state-dependent safety constraints such as obstacle avoidance, forbidden regions, and collision restrictions. The agent must complete goal-oriented objectives while avoiding unsafe states throughout the trajectory. Constraint violations incur safety costs or terminate the episode, discouraging unsafe exploration and emphasizing safety during both learning and execution.

**Velocity-Constrained Control Tasks.** We further consider MuJoCo-based tasks with explicit safety constraints on system behavior, where agents are required to satisfy predefined safe velocity limits during task execution.

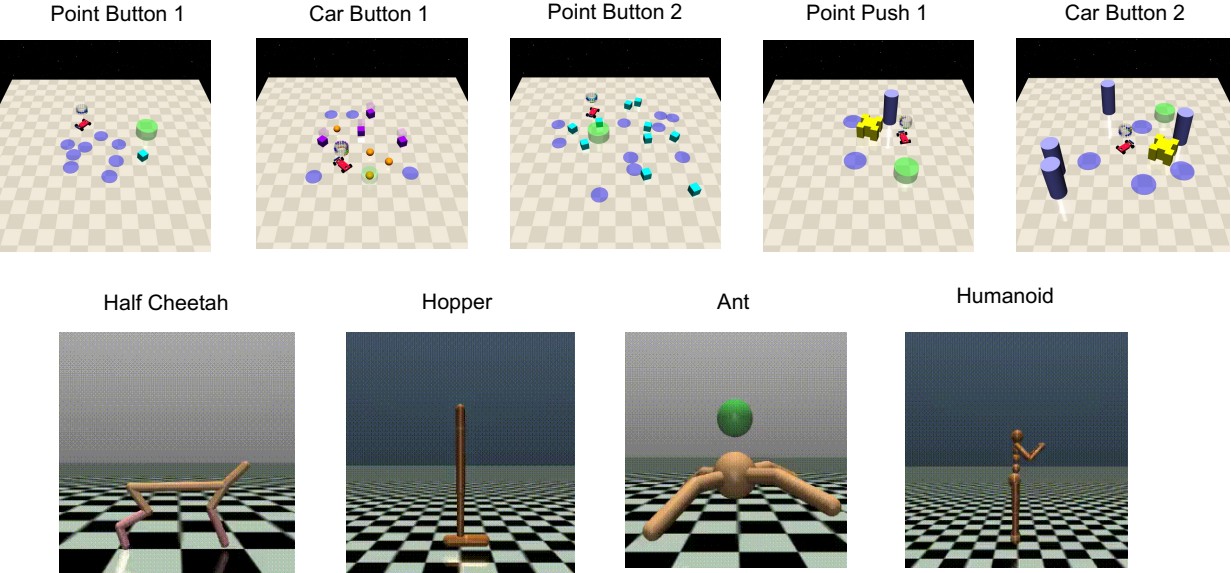

*Figure 5.* Task environments used in our experiments. Top row: Safe-Gym manipulation and navigation tasks, including Point Button, Car Button, and Point Push, which require the agent to accomplish goal-oriented behaviors while satisfying safety constraints such as obstacle avoidance and region constraints. Bottom row: Velocity-constrained MuJoCo locomotion tasks, including HalfCheetah, Hopper, Ant, and Humanoid, where agents must learn stable and efficient locomotion policies under explicit velocity limits and safety-related constraints.

## E.2. Network Architectures

*Table 3.* ALGD Structure Summary.

| Module | Sub-Module | Structure/Computation | Output Dimension |
|---|---|---|---|
| Diffusion Policy $\pi_\theta$ | Time Emb. | `Embedding`$(K, 16)$ | 16 |
| | Score Network | Input concatenation $[s, x_k, e(k)]$ | |
| | | `Linear`$(d_s + d_a + 16 \rightarrow 128)$ + ReLU | 128 |
| | | `Linear`$(128 \rightarrow 128)$ + ReLU | 128 |
| | | `Linear`$(128 \rightarrow 128)$ + ReLU | 128 |
| | | `Linear`$(128 \rightarrow d_a)$ | $d_a$ |
| | Schedule | $\sigma_t \sim$ `loglinspace`$(10^{-4}, 10^{-1})$, | |
| Reward Critic $Q_\psi(s, a)$ | $Q_1$ | `Linear`$(d_s + d_a \rightarrow 256)$ + ReLU | 256 |
| | | `Linear`$(256 \rightarrow 256)$ + ReLU | 256 |
| | | `Linear`$(256 \rightarrow 1)$ | 1 |
| | $Q_2$ | same with $Q_1$ (Double Q structure) | 1 |
| | Output | Forward Output $(Q_1, Q_2)$ | $(1, 1)$ |
| Safety Critic Ensemble $\bar{Q}_c \xi$ | Ensemble Size | $M =$ `qc_ens_size` (parallel ensemble) | $M$ |
| | Layer 1 | `EnsembleFC`$(d_s + d_a \rightarrow 256)$ + SiLU, weight_decay=$3 \times 10^{-5}$ | $M \times 256$ |
| | Layer 2 | `EnsembleFC`$(256 \rightarrow 256)$ + SiLU, weight_decay=$6 \times 10^{-5}$ | $M \times 256$ |
| | Layer 3 | `EnsembleFC`$(256 \rightarrow 1)$, weight_decay=$10^{-4}$ | $M \times 1$ |

## E.3. Training Hyperparameters

*Table 4.* Hyperparameters and configuration summary.

| Hyperparameter | Value | Hyperparameter | Value |
|---|---|---|---|
| discount $\gamma$ | 0.99 | safety discount $\gamma_c$ | 0.99 |
| reward critic hidden dim | 256 | safety critic hidden size | 256 |
| actor/critic lr | $3 \times 10^{-4}$ | safety critic lr | $3 \times 10^{-4}$ |
| replay buffer size | $10^6$ | num train repeat (per epoch) | 10 |
| Polyak averaging | 0.005 | frequency | 5 |
| policy train batch size | 256 | number of seeds | 10 |
| augmented lagrangian $\rho$ | 1.0 | safety critic ensemble size $M$ | 6 |
| diffusion steps $K$ | 5 | score network hidden dim | 128 |
| time embedding dim | 16 | Monte Carlo score estimation $N$ | 6 |

## E.4. Comparisons with On-policy Baselines

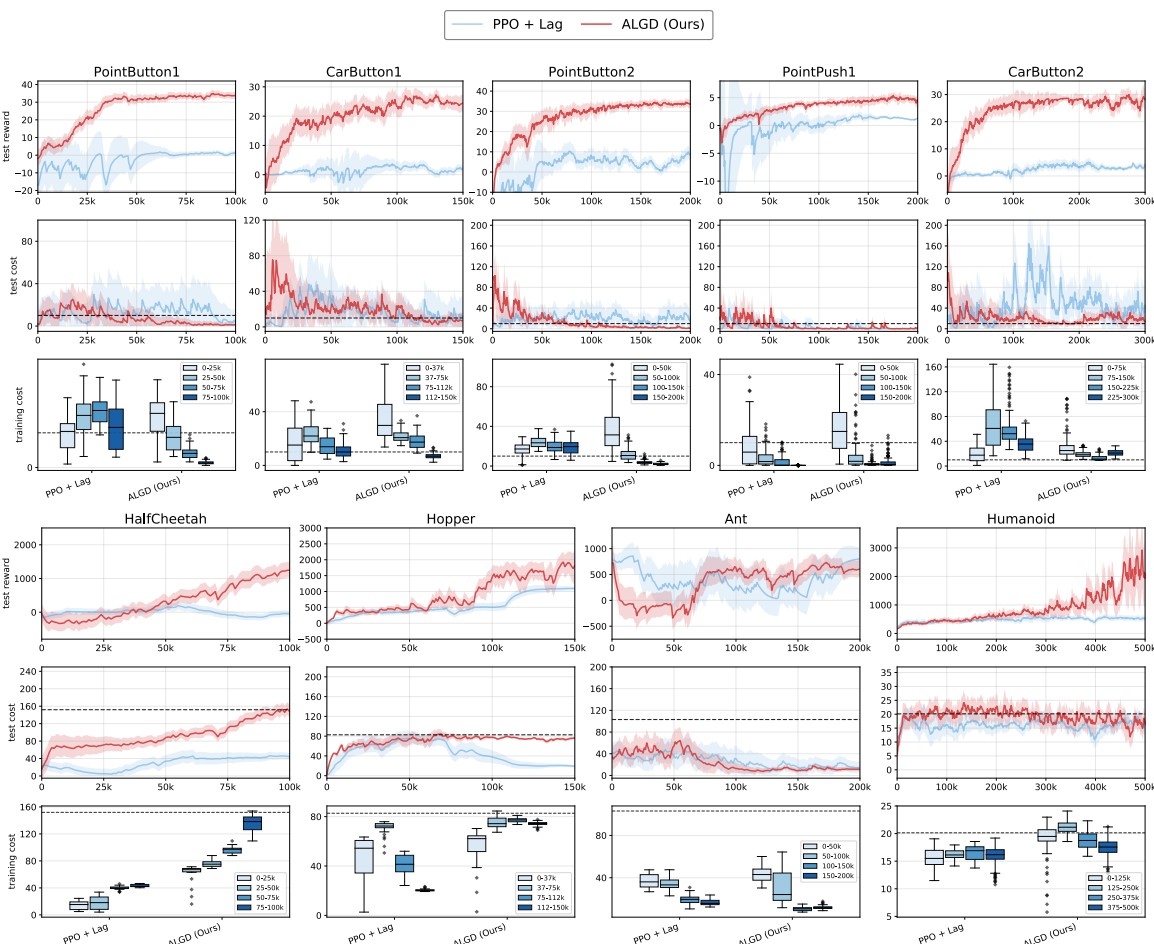

*Figure 6.* Comparisons with on-policy baselines on Safety-Gym (top half) and velocity-constrained MuJoCo (bottom half). In general, compared with the on-policy baselines, ALGD achieves competitive returns while demonstrating improved sample efficiency and stronger safety performance.

### E.5. More Ablation Study

**Monte Carlo Score Estimation.**  Figure 7 presents the full results of Monte Carlo score estimation.

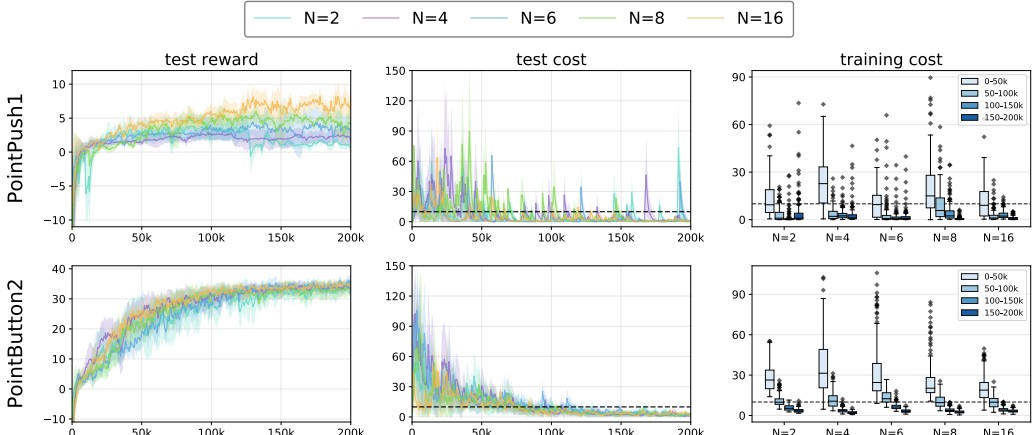

*Figure 7.* Ablation study on Monte Carlo score estimation. We evaluate different numbers of Monte Carlo samples $N \in \{2, 4, 6, 8, 16\}$ on PointPush1 (top row) and PointButton2 (bottom row). From left to right, we report the test reward, test cost, and training cost over the course of training. As $N$ increases, the learned policies consistently achieve higher test reward and lower test cost, while also exhibiting reduced training cost and variance. This monotonic improvement empirically validates the theoretical result: using more Monte Carlo samples yields a more accurate score estimation, leading to better performance–cost trade-offs and more stable learning.

**Ensemble Cost Critics.**  In the ablation study (shown in Figure 8), we observe that employing an ensemble of cost critics leads to more stable cost estimation, with substantially reduced variance throughout training. This improved stability translates into lower accumulated training cost and more reliable policy performance at test time. These results validate the effectiveness of our ensemble design: averaging over multiple critics provides a more accurate estimate of the augmented Lagrangian, bringing it closer to the underlying ground-truth objective. Importantly, the additional computational overhead introduced by the ensemble remains modest in practice. Since the critics can be evaluated in parallel on GPU architectures, increasing the ensemble size incurs only a marginal increase in per-step computation time, as confirmed by the runtime analysis in Table 5.

*Notably, our algorithm already exhibits stable learning and safe policy behavior even without the ensemble ($M = 1$); increasing the ensemble size does not alter the underlying energy landscape, but instead strengthens the approach by allowing $\bar{Q}_c$ to provide a more accurate estimate of $\nabla_a \mathcal{L}_A(s, a, \lambda)$, thereby improving gradient quality and overall algorithm performance.*

To rule out potential confounding effects, we evaluated the use of cost critic ensembles in the baseline methods, including SAC+Lag and CAL (originally proposed with ensembles). In fact, all baseline results reported in Figure 3 already employ the same cost-critic ensemble size ($M = 6$) as ALGD. Despite using the same cost-critic ensemble, these baselines remain consistently outperformed by ALGD across all tasks.

*Table 5.* Per-step GPU computation time (ms) for different critic ensemble sizes $M$, reported as mean $\pm$ standard deviation.

| **Task** | $M{=}1$ | $M{=}2$ | $M{=}4$ | $M{=}6$ | $M{=}8$ | $M{=}16$ |
|---|---|---|---|---|---|---|
| PointPush1 | $4.626 \pm 0.986$ | $4.779 \pm 1.015$ | $4.812 \pm 1.013$ | $4.863 \pm 1.030$ | $4.900 \pm 1.042$ | $5.048 \pm 1.053$ |
| PointButton2 | $4.039 \pm 0.973$ | $4.164 \pm 0.996$ | $4.191 \pm 1.002$ | $4.240 \pm 1.018$ | $4.358 \pm 1.020$ | $4.417 \pm 1.026$ |

**Degree of Convexification.**  We study the effect of the convexification strength $\rho$ in the augmented Lagrangian through an ablation analysis. Figure 9 reports the performance under different values of $\rho$, illustrating how the quadratic penalty term influences both optimization stability and exploration behavior.

Overall, we observe that a *moderate* range $\rho \in [0.5, 2]$ leads to the most stable learning dynamics. When $\rho$ is too small, the augmented Lagrangian provides insufficient convexification, and the resulting denoising dynamics remain unstable, leading

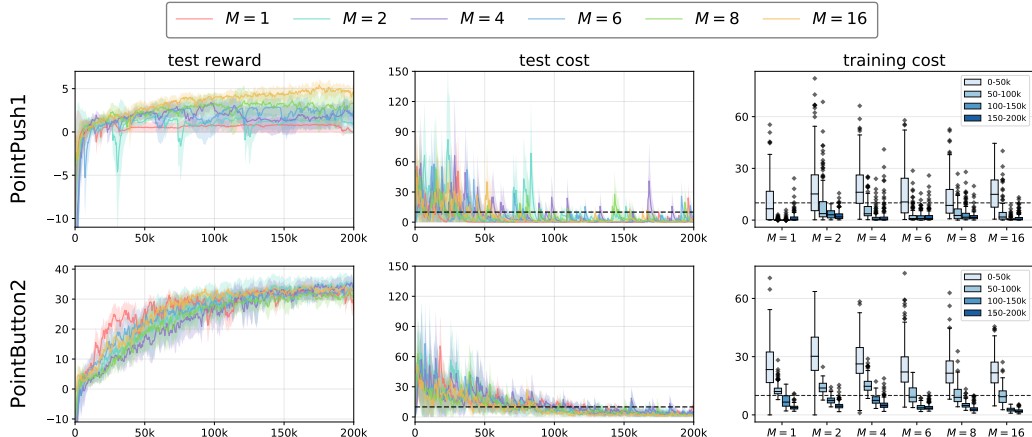

*Figure 8.* Ablation Studies of critic ensemble size $M$. We study the effect of varying the critic ensemble size $M \in \{1, 2, 4, 6, 8, 16\}$ on PointPush1 (top row) and PointButton2 (bottom row). From left to right, the plots show test reward, test cost, and training cost throughout training. Increasing $M$ consistently improves performance, yielding higher test rewards and lower test costs, while also reducing training cost and variance. These results indicate that larger critic ensembles provide more accurate and stable value estimation, leading to safer and more efficient policy learning, in line with our theoretical analysis.

to high variance in both reward and cost during training. In contrast, when $\rho$ becomes excessively large, the augmented Lagrangian $\mathcal{L}_A$ attains extreme values. As a consequence, during Monte Carlo score estimation, the Boltzmann weights $\pi_A(a|s) \propto \exp(-\frac{\mathcal{L}_A(s,a,\lambda)}{\beta})$ concentrate on a very small subset of the action space, causing probability mass to collapse into narrow regions. This over-concentration significantly reduces effective exploration and degrades learning performance.

These results highlight the importance of choosing an appropriate convexification strength: a moderate $\rho$ strikes a balance between stabilizing the optimization landscape and maintaining sufficient exploration, thereby yielding the best empirical performance.

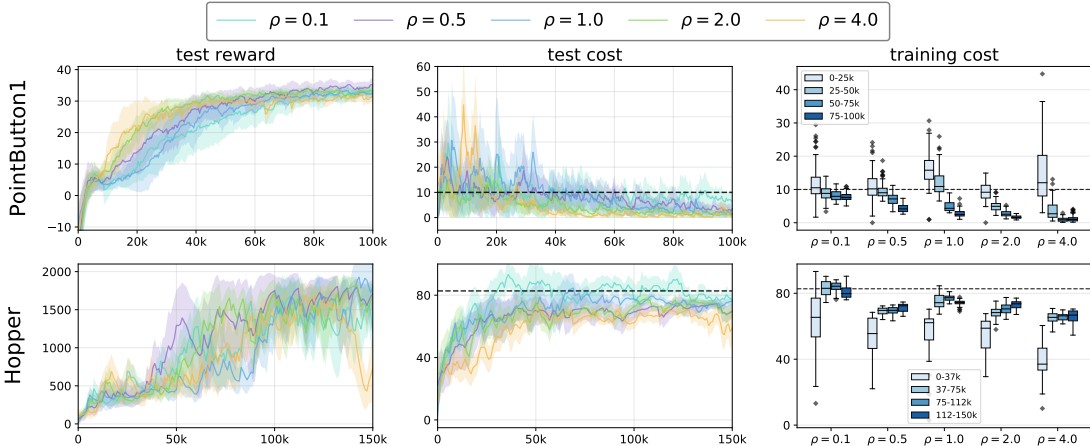

*Figure 9.* **Effect of the convexification strength $\rho$ in the augmented Lagrangian.** We report test reward (left), test cost (middle), and training cost statistics (right) on the PointButton1 task for different values of $\rho$. A moderate convexification strength leads to stable denoising dynamics, consistent cost satisfaction, and strong reward performance. When $\rho$ is too small, the augmented Lagrangian provides insufficient convexification, resulting in unstable dynamics and high variance. In contrast, excessively large $\rho$ yields extreme values of the augmented Lagrangian, causing Monte Carlo reweighting via $\exp(-\frac{\mathcal{L}_A(s,a,\lambda)}{\beta})$ to collapse the probability mass onto narrow regions of the action space, thereby reducing exploration and degrading performance.

## E.6. More Augmented Lagrangian Compare Results

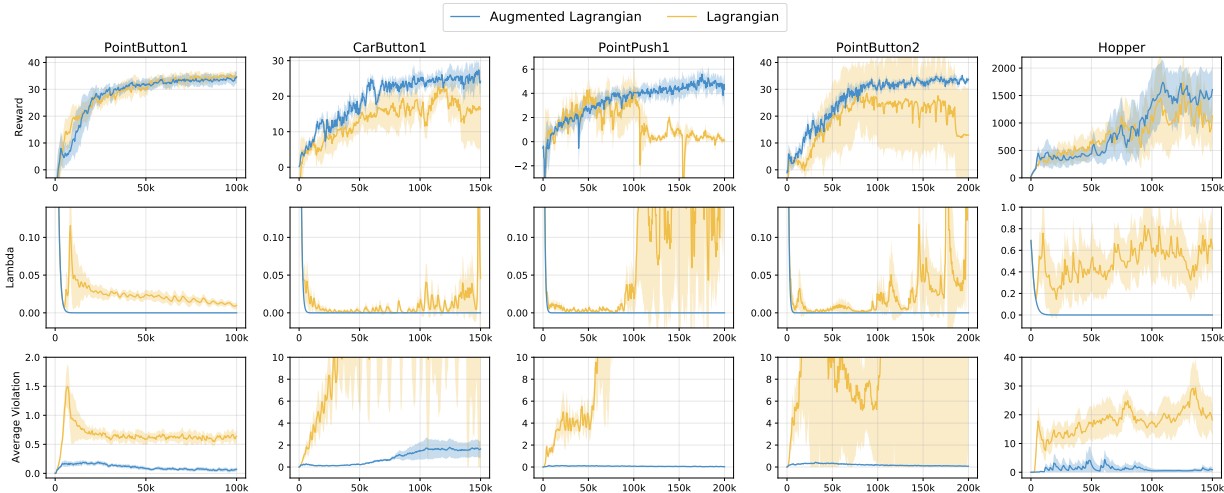

*Figure 10.* Full comparative analysis of training stability between the standard Lagrangian (yellow) and the augmented Lagrangian (blue). (Top) Evaluation rewards; (Middle) Dual variable $\lambda$ updates; (Bottom) Average constraint violation (calculated as $\max(0, c(s) - h)$). The augmented formulation directly addresses the oscillation of dual variables (L2) and the instability of the induced Boltzmann distribution (L3) by regularizing the local curvature of the energy landscape. This results in more stable denoising dynamics and dual updating, allowing the policy to reach the safety threshold significantly faster and with zero dual variables.

## E.7. Wall-Clock Time Comparison

*Table 6.* Runtime statistics (mean $\pm$ std, seconds) on Safety Gym and MuJoCo. ALGD incurs a longer training time due to the cost of Monte Carlo–based score estimation during training. However, at evaluation time, the learned score function is directly used, amortizing most of the computational overhead, so the evaluation time remains comparable to other methods.

| Environment | Algorithm | Training time (s) | Eval time (s) |
|---|---|---|---|
| Safety-Gym | **ALGD (Ours)** | $28.60 \pm 0.26$ | $4.924 \pm 0.159$ |
| | CAL | $11.19 \pm 0.35$ | $2.914 \pm 0.097$ |
| | SAC+Lag | $10.48 \pm 0.39$ | $2.839 \pm 0.176$ |
| | SAC+AugLag | $10.06 \pm 0.23$ | $2.648 \pm 0.109$ |
| | PPO+Lag | $7.25 \pm 0.29$ | $1.915 \pm 0.285$ |
| | HJ | $9.51 \pm 0.29$ | $2.369 \pm 0.266$ |
| MuJoCo | **ALGD (Ours)** | $67.37 \pm 2.60$ | $0.651 \pm 0.093$ |
| | CAL | $25.82 \pm 0.60$ | $0.656 \pm 0.762$ |
| | SAC+Lag | $23.88 \pm 0.72$ | $0.341 \pm 0.166$ |
| | SAC+AugLag | $25.32 \pm 1.01$ | $0.380 \pm 0.096$ |
| | PPO+Lag | $18.57 \pm 0.16$ | $0.280 \pm 0.014$ |
| | HJ | $23.32 \pm 0.83$ | $0.620 \pm 0.845$ |

