# OpenReview forum: "How Does the Lagrangian Guide Safe Reinforcement Learning through Diffusion Models?"
_ICML.cc/2026/Conference — ICML 2026 regular_

### Official Review · Reviewer_cEtQ · 2026-02-17

**Soundness:** 3
**Presentation:** 3
**Significance:** 3
**Originality:** 3
**Overall Recommendation:** 5
**Confidence:** 4

**Summary:**

This paper proposes Augmented Lagrangian-Guided Diffusion, which combines the diffusion-based policy generation with augmented Lagrangian optimization.  The key insight is that directly using the standard Lagrangian as an energy function to guide diffusion-based policy sampling leads to unstable score fields due to the nonconvex energy landscape. To deal with this issue, the authors investigated an augmented Lagrangian, which locally convexifies the energy landscape near constraint boundaries, stabilizing denoising dynamics without altering the optimal policy distribution. Theoretical proofs are provided for the Lagrangian-guided score function derivation, local convexification, and final bound. Experiments on Mojoco and Gym empirically shows the effect of the author's algorithm on training stability and safety.

**Compliance With Llm Reviewing Policy:**

Affirmed.

**Final Justification:**

Although the authors did not fully solved my concern in the first rebuttal, the Reply Rebuttal Comment seems good and solve most of my questions. As I said, I think the paper may exceed 4.5, so I will change my score to 5.

**Key Questions For Authors:**

1. The discussion of the gap between the continuous case and the discrete case: Theorem 3.3 's bound bases on the Girsanov of the continuous path, which has some gap with the algorithm with K diffusion steps.

Recommendation: Discuss discretization error. Reference existing analyses from the score-based generative modeling literature (e.g., convergence guarantees for discretized reverse SDEs in Song et al., 2020 and subsequent work) to at least informally characterize the gap introduced by using K=5 discrete steps instead of the continuous-time process assumed by the theory. Even an informal discussion would be substantially better than the current complete omission.

2. The bound of $\epsilon = |L_A - L|$ requires $\lambda \to \lambda^*$. While in off-policy sampling, and diffusion-based policy generation, no convergence analysis is provided. The authors themselves acknowledge this in Appendix B.

Suggestions: If theoretically proving it is difficult, I suggest empirically track $\epsilon$ during training. Record the trajectory of $\|L_A(s,a,λ_k) - L(s,a,λ_k)\|$ and the dual variable λ_k across training iterations.

3. Theorem 3.2(a) only guarantees local convexification in a neighborhood of the constraint boundary where $|Q_c(s,a) - h|$ is "sufficiently small," but the size of this neighborhood is never characterized in terms of problem-dependent quantities (e.g., $\|\nabla_a Q_c\|$, $\|\nabla^2_a Q_c\|$, $\rho$). This makes it impossible to assess whether the convexification effect is practically meaningful or essentially vacuous in typical safe RL tasks.

Suggestions: Provide convergence guarantees in a simplified setting. Even if a full analysis under neural function approximation is intractable, convergence results under tabular or linear function approximation would significantly strengthen the theoretical contribution. At minimum, state sufficient conditions on the learning rates and penalty parameter $\rho$ under which the primal-dual updates converge, analogous to classical results in the augmented Lagrangian literature.

**Limitations:**

Yes

**Strengths And Weaknesses:**

Soundness:

Strength: The theoretical proof is coherent. Proposition 3.1 derives the exact score function under the Lagrangian energy, Theorem 3.2 proves local convexification and optimal policy invariance, and Theorem 3.3 provides a finite-sample KL divergence bound.

 Weakness:

Theorem 3.2(a) only guarantees local convexification when the constraint is active and near the boundary. In practice, I think that $Q_c$ is trained by a neural network, so how do we ensure the bound of $|Q_c - h|$? To show the "where the dominant additional term is positive semidefinite.", we may require a boundary of $|Q_c-h|$ with the Lipschitz, or Hessian of $Q_c$?

The Theorem 3.3 gives a bound on $\epsilon = |L_A - L|$. However, this $\epsilon$ is never bounded in terms of algorithm quantities, like the number of iterations, learning rate, etc. So what should be the convergence rate for the algorithm?

In Lemma D.1, there is a sub-gaussian assumption, but it is not verified for the neural network function approximators used in practice.

Presentation:

Strength: The paper is well-organized with a clear narrative: limitation identification (L1-L3) → theoretical connection → augmented solution → algorithm → experiments.

Figure 1 provides excellent intuition for the energy landscape difference between standard and augmented Lagrangian.
The Q&A-style result analysis (Questions 1-4) in Section 4 is effective for guiding the reader.

Weakness: The log-sum-exp trick seems to be an important part for reproduction. However, it is just discussed in the Appendix. What's more, the Algorithm itself is also put on an inconspicuous place of the paper. Maybe the authors may change a little bit of the structure.

The relationship between the discrete algorithm (Algorithm 1) and the continuous-time theoretical analysis could be made more explicit. The discretization error of the reverse-time SDE is not discussed.

Significance:
Strengths:
The paper addresses an important and underexplored problem: online safe RL with expressive policy representations. The combination of diffusion models with constrained optimization is timely and relevant.

The empirical results are comprehensive, covering 9 tasks across two benchmarks, with ablation studies on all key hyperparameters (N, M, ρ).

The observation that naively combining augmented Lagrangian with SAC (Gaussian policy) does not help, while ALGD succeeds, highlights the importance of the diffusion policy representation.

Weaknesses:

The training time overhead is substantial (~2.5-3x compared to baselines, Table 6). While evaluation time is comparable, the training cost may limit practical applicability.

All experiments are in simulation. The paper acknowledges this limitation but does not discuss how factors like real-time constraints or model mismatch might affect the approach.

The improvement over CAL (which also uses augmented Lagrangian ideas) varies across tasks, and in some cases (e.g., Humanoid reward), the margins are modest.

Originality:
Strengths:
The interpretation of the Lagrangian as an energy function guiding diffusion denoising, and the identification that this creates instability, is a novel and insightful contribution.

The connection between augmented Lagrangian and local convexification of the diffusion energy landscape is original and bridges optimization theory with generative modeling in a non-trivial way.

The Monte Carlo score estimation with importance weighting from the augmented Lagrangian (Eq. 11) is a practical and well-motivated design choice.

Weaknesses:
The ensemble cost critic design is standard and has been used in prior work (e.g., CAL). The paper acknowledges this but could better distinguish its contribution.

---

> ### Author Rebuttal · Authors · 2026-03-29
>
> We thank the reviewer for the feedback and recognition.
>
> ---
>
> **Soundness W1 and Q3**  `convexification and boundedness.`
>
> We thank the reviewer for this insightful question.
>
> Local convexification stabilizes denoising by reducing oscillations at safety boundaries where gradient instability is most pronounced. Without it, the irregular landscape makes score matching unstable, preventing the diffusion from matching the target Boltzmann distribution. This leads to distribution mismatch and oscillating dual variables, blocking the Lagrangian from reaching equilibrium (Figure 10).
>
> **Boundedness.** Since benchmark cost functions are continuous, the state–action spaces are compact, and the learned critic $Q_c$ is a continuous function (due to the smooth SiLU activation) on the compact domain, $|Q_c - h|$ remains bounded. Moreover, we employ weight decay and ensemble critics, which regularize sharp gradients. Therefore, the assumptions of local boundedness on $ \| \nabla_a Q_c \| $ and Hessian of $Q_c$ are mild. Under these conditions, existing $\rho > 0$ can convexify a neighborhood of safety boundaries, ensuring positive curvature.
>
> **Empirical Test.**  We provide empirical test by sampling in compact domain. The result shows bounded means and variances, suggesting that the smoothness are satisfied.
>
> |envs|$\| \nabla_aQ_c \|$ |$tr(\nabla_a^2Q_c)$|
> |-|-|-|
> |PointButton2|0.52±0.61|0.56±1.01|
> |PointPush1|0.56±0.38|0.40±0.95|
>
> **Soundness  W2 and Q2** `\epsilon gap.`
>
> Thanks for your question. Deriving convergence rates for constrained deep RL is an open challenge, we instead give an asymptotic convergence analysis by a two-stage logic:
>
> (1) **Objective Invariance.** According to Theorem 3.2, the augmented Lagrangian $L_A$ preserves both the optimal value and the optimal policy distribution of the original problem $L$ at the saddle point. This ensures that the local convexification ($\rho > 0$) used to stabilize training does not alter the optimum.
>
> (2) **Convergence of the Primal-Dual.**  Following established results in safe RL (e.g., RCPO [1], Theorem 2), the primal-dual update converges to the equilibrium of $\pi$ and $\lambda$, under sufficient conditions such as multi-timescale learning rates ([1], Assumption 3). As augmented form moves the saddle point, the penalty term in $L_A$ vanishes due to Objective Invariance (1), leading to $\epsilon \to 0$ as the iteration step to infinity.
>
> [1] Chen et al. "Reward constrained policy optimization.".
>
> **Connect to Theorem 3.3.** The distribution discrepancy is bounded by combining of the optimization gap (shrinks with iterations) and the Monte Carlo error (shrinks with $N$).
>
> **Empirical verification.** We provide new empirical tracking of the gap $\epsilon$ and the dual variable $\lambda$ across training steps (see table below), which indicates the convergence to equilibrium.
>
> - PointButton2
>
> |step|$\|L_A - L\|$|$\lambda$|
> |-|-|-|
> |50k|0.34±0.03|0.27±0.02|
> |100k|0.11±0.01|0.03±0.02|
> |200k|0.01±0.00|0.00±0.00|
>
> - PointPush1
>
> |step|$\|L_A - L\|$|$\lambda$|
> |-|-|-|
> |50k|0.28±0.02| 0.10±0.01 |
> |100k|0.12±0.01| 0.02±0.02 |
> |200k|0.01±0.00| 0.00±0.00 |
>
> **Soundness W3** `sub-Gaussian.`  The assumption is commonly used (see Lines 1097–1099) in Monte Carlo sampling. In practice, it can be relaxed to weaker conditions (e.g., bounded higher-order moments) without affecting the overall analysis.
>
> **Presentation W1** `log-sum-exp trick.` We agree and will move the trick and Algorithm 1 to the main text to improve clarity.
>
> **Presentation W2 and Q1** `discretization error.` We will clarify in the new version that the discrete-time sampling introduces a controllable discretization error, which can be incorporated into Theorem 3.3.
>
> **Significance  W1** `training time.` Higher training cost is inherent in diffusion policy. However, ALGD supports amortized inference, leading to fast evaluation. Also, training efficiency can be improved by JAX.
>
> **Significance  W2** `sim-to-real.` We agree this limitation in RL community. In practice, ALGD supports real-time deployment through policy distillation, and sim-to-real transfer can be addressed via fine-tuning to adapt to model mismatch.
>
> **Significance W3** `humanoid reward.` On Humanoid task, ALGD shows advantages with longer training, achieving higher rewards and lower costs (see table below). We will clarify in the next revision.
>
> | |Humanoid|500k|1M|
> |-|-|-|-|
> |CAL|Reward(Cost)|1315±650(22±4)|4827±2425(20±5)|
> |ALGD|Reward(Cost)|3081±492(18±5)|5237±1630(15±6)|
>
> **Originality W1** `ensemble cost critic.` It is a practical design for improving gradient stability, rather than a core contribution of our work indicated in Lines 67–85.
>
> **Q3** `convergence guarantees and sufficient conditions.` Thanks for valuable advice. The asymptotic convergence needs key conditions:
>
> - Multi-timescale learning rate assumption in [1].
> - Penalty Parameter $\rho > 0$ in Theorem 3.2.
>
> See proof in `Soundness W2` and Convexification in `Soundness W1`.

---

> > ### Author Rebuttal · Reviewer_cEtQ · 2026-04-01
> >
> > I thank the authors for the detailed rebuttal. The empirical tracking of ε and λ convergence is convincing and the two-stage asymptotic argument via RCPO is a reasonable strategy given the difficulty of finite-time analysis in deep RL. The extended Humanoid results at 1M steps also strengthen the empirical contribution. I may raise my score to 4.5. To further increase to 5, we ask the authors to address the following:
> > 1. The authors provide the boundness of $|Q_c - h|$, but as we have shown in equation (10), the authors still need to provide why the $\rho \nabla_a Q_\pi^c (\nabla_a Q_\pi^c)^⊤$ is really the dominant factor (or it actually is not?).
> > 2. about the discretization error revision, Chen et al. (2023, "Sampling is as easy as learning the score")'s framework might be a feasible way. However, it might require $K = Õ(d/ε²)$, which might not be fit for the $K=5$ case. An another way is that adding a bound in $\epsilon_disc(K, d, L_score)$ after Thm 3.3, and argue that it is controllable in RL case. Anyway, I am looking forward to the authors' revision about that.
> >
> > Overall, I think the authors have solved most of my concern, so I keep my positive score. To raise my score to 5, more analysis of the statement in equation 10 is required.

---

> > > ### Author Response · Authors · 2026-04-02
> > >
> > > **Q1** `dominance verification.` Thanks for your question. We clarify that the dominance of the term $\rho \nabla_a Q_c (\nabla_a Q_c)^\top$ is localized near the boundary, and holds in a spectral sense.
> > >
> > > **Theoretical Perspective.**
> > >
> > > 1. **Mathematical Dominance:** From Eq.10, the Hessian of the augmented objective is:  $$\nabla_a^2 L_A = \nabla_a^2 L + \rho (Q_c - h) \nabla^2_a Q_c + \rho \nabla_a Q_c (\nabla_a Q_c)^\top . $$
> > > Near the boundary, where $|Q_c - h|$ is small, the second term becomes negligible, making the rank-1 positive semidefinite term the primary contributor to curvature in this region. Therefore, the dominance holds in a spectral sense when the term $ \rho \nabla_a Q_c \nabla_a Q_c^\top $ is strong enough to offset the negative eigenvalues of $\nabla_a^2 L + \rho (Q_c - h) \nabla^2_a Q_c $.
> > >
> > > 2. **Controlled Regularization:** We avoid global dominance of $\rho$ to prevent policy collapse (see Lines 1671-1677), ensuring instead that convexification occurs only near the boundary (where the gradient is highly oscillating), where it stabilizes optimization without suppressing multi-modality.
> > >
> > > **Empirical Verification:** To verify our hypothesis, we conducted spectral analysis of the Hessian eigenvalues ($\sigma$) of both the standard Lagrangian ($L$) and the augmented Lagrangian ($L_A$) by varying $\rho \in [0.1, 4.0]$. Please note, the exact values of $|Q_c - h|$ vary due to the stochastic nature of sampling (indicated by the standard deviations), the trend remains robust: the penalty term effectively overcomes the Hessian's minimal eigenvalue in the critical boundary regions.
> > >
> > > 1. **Spectral Transition.** In the "near-boundary" region, the original Lagrangian is non-convex ($\sigma_{\min}(\nabla_a^2 L) < 0$). However, with $\rho \in [0.5, 2.0]$, the term $\rho \nabla_a Q_c (\nabla_a Q_c)^\top$ (with $\sigma_{\max} \approx 0.3-0.7$) successfully corrects this negative curvature, resulting in a positive definite Hessian ($\sigma_{\min}(\nabla_a^2 L_A)>0$).
> > >
> > > 2. **Local Convexification.** The dominance is effectively localized near the boundary, where negative curvature arises. Away from the boundary, the objective is generally well-conditioned, so the convexification term is not essential, although it may still increase curvature. Empirically, small $\rho$ fails to overcome negative curvature, while large $\rho \to 4$ leads to reduced multi-modality, indicating over-regularization. We find $[0.5, 2.0]$ achieves the balance, without sacrificing expressiveness (aligned with Figure 9 in Appendix).
> > >
> > > **PointButton1**
> > >
> > > - $\rho = 0.1$
> > >
> > > |region|Multi-modal Rate (%)|$\|Q_c - h\|$|$\sigma_{max}(\rho\nabla_a Q_c \nabla_a Q_c^\top)$| $\sigma_{min} (\nabla_a^2 L)$ | $\sigma_{min} (\nabla_a^2 L_A)$|
> > > |-|-|-|-|-|-|
> > > |near|59.2± 10.6|0.026 ± 0.017|0.109 ± 0.118|-0.148±0.118|-0.105±0.124|
> > > |away|-| 2.048 ± 0.347|0.147 ± 0.091|0.031±0.042|0.029±0.084|
> > >
> > > - $\rho = 0.5$
> > >
> > > |region|Multi-modal Rate (%)|$\|Q_c - h\|$|$\sigma_{max}(\rho\nabla_a Q_c \nabla_a Q_c^\top)$| $\sigma_{min} (\nabla_a^2 L)$ | $\sigma_{min} (\nabla_a^2 L_A)$|
> > > |-|-|-|-|-|-|
> > > |near| 61.8± 13.8|0.028±0.027| 0.285±0.189| -0.139±0.151|0.042±0.141|
> > > |away| - |1.905±0.512|0.287±0.231|0.029±0.037|0.086±0.073|
> > >
> > >
> > > - $\rho = 1.0$
> > >
> > > |region|Multi-modal Rate (%)|$\|Q_c - h\|$|$\sigma_{max}(\rho\nabla_a Q_c \nabla_a Q_c^\top)$| $\sigma_{min} (\nabla_a^2 L)$ | $\sigma_{min} (\nabla_a^2 L_A)$|
> > > |-|-|-|-|-|-|
> > > |near|58.1±6.3|0.033±0.030|0.577±0.273| -0.158±0.124|0.077±0.085|
> > > |away|-|1.779±0.156|0.653±0.142|-0.014±0.034|0.095±0.078|
> > >
> > >
> > > - $\rho = 2.0$
> > >
> > > |region|Multi-modal Rate (%)|$\|Q_c - h\|$|$\sigma_{max}(\rho\nabla_a Q_c \nabla_a Q_c^\top)$| $\sigma_{min} (\nabla_a^2 L)$ | $\sigma_{min} (\nabla_a^2 L_A)$|
> > > |-|-|-|-|-|-|
> > > |near|48.1±5.1| 0.096±0.138|0.706±0.146|-0.133±0.192|0.092±0.131|
> > > |away|-| 1.999±0.218|0.871±0.245|0.011±0.027|0.133±0.104|
> > >
> > > - $\rho = 4.0$
> > >
> > > |region|Multi-modal Rate (%)|$\|Q_c - h\|$|$\sigma_{max}(\rho\nabla_a Q_c \nabla_a Q_c^\top)$| $\sigma_{min} (\nabla_a^2 L)$ | $\sigma_{min} (\nabla_a^2 L_A)$|
> > > |-|-|-|-|-|-|
> > > |near|29.5±14.2|0.027±0.016|1.261±0.285|-0.164±0.161|0.224±0.110|
> > > |away|-| 2.159±0.335|1.374±0.046| -0.024±0.046|0.273±0.131|
> > >
> > > **Q2** `discretization error.`  We thank the reviewer for the suggestion. We adopt the second approach (for $K=5$) by explicitly incorporating a discretization error term $\varepsilon (K, m, L_s)$ into our main theorem, where $K, m$ and $L_s$ denote the number of denoising steps, the action dimension, and the Lipschitz constant of the score function, respectively. By leveraging results from [1], we show that this error term admits a polynomial decay with respect to $K$, and is therefore controlled in our setting.
> > >
> > >  [1] Chen, Sitan, et al. "Sampling is as easy as learning the score: theory for diffusion models with minimal data assumptions.".
> > >
> > >
> > > ----
> > > We will incorporate all the above results into the camera-ready version. We hope these additions address the reviewer’s concerns and help improve the overall evaluation.

---

### Official Review · Reviewer_igX2 · 2026-03-11

**Soundness:** 3
**Presentation:** 3
**Significance:** 3
**Originality:** 3
**Overall Recommendation:** 4
**Confidence:** 3

**Summary:**

This paper studies safe reinforcement learning with diffusion-based policies from the perspective of Lagrangian guidance. The authors argue that when diffusion models are used as policies, the Lagrangian constraint effectively shapes the energy landscape guiding the denoising process. Based on this observation, the paper proposes an augmented-Lagrangian-guided diffusion policy framework to improve the stability of constrained policy optimization. The paper provides theoretical intuition linking augmented Lagrangian geometry with diffusion guidance and evaluates the method on several safe control benchmarks, showing improved reward–cost trade-offs.

**Compliance With Llm Reviewing Policy:**

Affirmed.

**Final Justification:**

After reading all the rebuttals, I keep my current evaluation.

**Key Questions For Authors:**

1.	Can the authors provide additional analysis demonstrating when diffusion policies offer clear advantages in the evaluated safe RL tasks? In particular, do the environments exhibit multi-modal action distributions or constraint-induced multi-modal behavior?
2.	How do the learned diffusion policies behave near safety boundaries? Do they exhibit multiple feasible action modes in such regions?
3.	Given that the practical algorithm uses stochastic optimization and approximate critics, how should readers interpret the augmented-Lagrangian analysis? Which aspects of the theoretical intuition are expected to carry over to the practical training dynamics?
4.	Could the authors clarify the exact cost limits used in each environment and report them explicitly for reproducibility?

**Limitations:**

yes

**Strengths And Weaknesses:**

Strengths:
1. The paper offers a novel interpretation of Lagrangian guidance in diffusion-based policies by viewing the constraint term as part of the energy function guiding the denoising process. This perspective provides a useful conceptual bridge between constrained RL, energy-based modeling, and diffusion policies. Even though augmented Lagrangian is a classical optimization technique, applying it in this context and analyzing its interaction with diffusion policy learning is a creative and meaningful contribution.
2. The algorithm design follows naturally from the proposed interpretation of the energy landscape, and the empirical results generally support the claims that the method can improve reward–cost trade-offs and training stability on the evaluated benchmarks. The paper is also clearly written and the overall narrative is easy to follow.

Weaknesses:
1. One aspect that could be further strengthened is the empirical justification for using diffusion policies in these safe RL environments. Diffusion policies are often motivated by their ability to represent multi-modal action distributions. However, it is not entirely clear whether the benchmark tasks used in the paper actually exhibit such multi-modality.
2. The theoretical discussion is interesting and provides intuition for why augmented Lagrangian guidance may improve the energy landscape. However, the practical algorithm relies on stochastic optimization, approximate critics, and diffusion-based policy learning. It would be helpful for the authors to clarify how the theoretical intuition should be interpreted under these approximations.
3. Minor clarity issues.
Some experimental details could be reported more explicitly to improve reproducibility. For example, the exact cost limits used in each environment should ideally be stated clearly rather than only indicated implicitly in figures.

---

> ### Author Rebuttal · Authors · 2026-03-28
>
> Thank you for your insightful and encouraging feedback, we truly appreciate your recognition of our contributions.
>
> ---
>
> **weakness 1 and Question 1** `benchmark multi-modality.`
>
> We thank the reviewer for this question. In Safety-Gym tasks [1] (e.g., PointButton, CarButton), obstacles block direct paths, creating multiple distinct safe routes and thus disconnected feasible action regions, which induce multi-modal policies (supported by `Empirical Evidence Question 2`). Gaussian policies collapse these modes via averaging, often producing unsafe actions. Consistently, replacing diffusion with a Gaussian policy under the same framework leads to clear performance degradation (Lines 409–431), confirming that modeling multi-modality is essential.
>
> [1] Safety-Gymnasium. https://safety-gymnasium.readthedocs.io/en/latest/index.html
>
> **weakness 2 and Question 3** ` clarify the theoretical intuition.`
>
> We thank the reviewer for the question. Our theory is not intended to guarantee exact optimality under approximations, but to provide a structural improvement of the energy landscape and policy learning. The augmented Lagrangian introduces additional positive curvature, which remains effective under approximate critics (trained via stochastic optimization) and helps stabilize score estimation.  To bridge theory and practice, we employ an ensemble critics to improve gradient accuracy. In addition, Theorem 3.3 establishes a finite-sample bound showing that the learned policy is close to the target Boltzmann distribution up to a controlled approximation error. Empirically, this leads to significantly more stable training and reduced dual oscillations (Figure 2).
>
> **weakness 3 and Question 4** `exact cost limits.`
>
> We thank the reviewer for the suggestion. The cost limits used in experiments follow the default settings in CAL:
> - Safety-Gym environments: 10.000
> - Ant: 103.115
> - HalfCheetah: 151.989
> - Hopper: 82.748
> - Humanoid: 20.140
>
> We will include these details in the camera-ready version.
>
> **Question 1** `additional analysis for clear advantages in safe RL tasks?`
>
> We thank the reviewer for this question. Diffusion policies are particularly beneficial when safety constraints induce non-convex and disconnected feasible action regions, leading to inherently multi-modal optimal policies. We verify this via additional ablations: under the same augmented Lagrangian framework, replacing the diffusion policy with a Gaussian one (SAC-AugLag) leads to unstable performance and higher constraint violations, especially in dynamic environments like CarButton (with multiple moving obstacles).
>
> - PointButton2
>
> | Method| Metric| 25% steps | 50% steps| 75% steps| 100% steps |
> |-|-|-|-|-|-|
> | ALGD | Reward ↑ (Cost ↓) | 26.18±8.60 (18.47±5.53)  | 35.34±3.62 (2.45±7.21)  | 35.79±3.12 (1.86±2.59)   | 35.48±3.63 (1.34±3.85)  |
> | SAC-AugLag | Reward (Cost) | 32.93±7.24 (26.56±15.16) | 39.23±6.92 (25.24±7.48) | 26.21±10.36 (30.2±12.40) | 20.63±5.26 (29.37±9.71) |
>
> - CarButton2
>
> | Method| Metric|25% steps |50% steps|75% steps|100% steps|
> |-|-|-|-|-|-|
> |ALGD| Reward (Cost) | 24.94±3.87 (19.52±11.64)|28.53±0.97 (10.18±1.74) | 29.81±3.46 (14.40±5.17) | 30.18±5.96 (9.29±4.26) |
> |SAC-AugLag|Reward (Cost)|18.08±5.32 (26.23±13.67) |26.76±6.46 (14.92±9.86) |4.45±7.53 (54.83±45.89)|1.88±2.45 (10.36±16.22)|
>
> **Question 2** `behave near safety boundaries? Do they exhibit multiple feasible action modes in such regions?`
>
> We thank the reviewer for this insightful question. Our analysis confirms that ALGD effectively stabilizes boundary behavior while preserving the expressive advantages of diffusion:
>
> **Behavior near safety boundaries.**  To make the near boundary behaviors more clear,  we monitor the gradient norm $\lVert \nabla_a \mathcal{L}_A(s, a^0, \lambda) \rVert $ at the final denoising step ($\tau=0$) . For ALGD with locally convexified landscape, the norm consistently approaches zero, indicating stable guidance toward constraint-satisfying KKT points (lower gradient norm). In contrast, standard Lagrangian-guided diffusion (LGD) exhibits oscillatory gradients, reflecting unstable denoising under irregular energy landscapes. This is also consistent with the observed stability of sampled actions near the boundary (also shown in Figure 10 in our paper).
>
> **More Empirical Evidence of Multi-modality.**   We quantify multi-modality via the percentage of safe boundary states where multiple distinct action modes are observed (identified from sampled actions). As shown below, ALGD maintains strong multi-modal behavior while achieving stable denoising dynamics.
>
> | Task | Final Denosing Gradient Norm ↓ | Multi-modal Rate at Boundary ↑ |
> | - |- | - |
> | PointButton2 | 0.059 ± 0.017 (vs. 0.121 ± 0.058 LGD) | 62.00% ± 2.00% |
> | PointPush1 | 0.056 ± 0.032 (vs. 0.118 ± 0.093 LGD) | 53.33% ± 9.14% |
>
> These findings demonstrate that the augmented Lagrangian provides a better-conditioned energy landscape, without suffering from mode collapse or training instability.

---

> > ### Author Rebuttal · Reviewer_igX2 · 2026-04-04
> >
> > I am satisfied with the rebuttal. I will keep my positive evaluation.

---

### Official Review · Reviewer_NCcb · 2026-03-13

**Soundness:** 3
**Presentation:** 4
**Significance:** 3
**Originality:** 3
**Overall Recommendation:** 5
**Confidence:** 3

**Summary:**

The paper proposes an off-policy algorithm to use diffusion policy for constrained reinforcement learning. The paper address the problem of instability in training for primal-dual methods for constrained RL. The paper claims that diffusion policy is well suited to handle the training instability of primal dual method because of its ability to represent multi-model policy. Further, the proposed method uses augmented Lagrangian to define the energy function for the diffusion policy and obtains smoother energy landscape compared to using just Lagrangian. The use of augmented Lagrangian stabilizes the training and smoother reward and cost performance curves are obtained for different tasks Safety-Gym benchmark and velocity constrained MuJoCo benchmark.

**Compliance With Llm Reviewing Policy:**

Affirmed.

**Final Justification:**

I am satisfied with the responses of the author, and I will maintain my score.

**Key Questions For Authors:**

1. Why is comparison with PPO-Lag not shown with off-policy methods?

**Limitations:**

Yes

**Strengths And Weaknesses:**

Strengths:
1. The paper is very well written. The problem statement and the solution to the problem is presented in a structured manner.
2. Visualization of energy landscape of augmented Lagrangian clearly shows how augmented lagrangian makes the energy landscape smooth. This strengthens the claims of the paper regarding stabilizing training.
3. Comparison with several off-policy safe RL algorithms has been shown and reward curves for the proposed algorithm are more stable.


Weaknesses:
1. The proposed algorithm doesn't always get best reward performance compared to the baselines.
2. Convergence guarantee for the proposed algorithm is not presented.
3. In Theorem 3.2, property (b), please shift comma outside the expectation.

---

> ### Author Rebuttal · Authors · 2026-03-28
>
> We thank the reviewer for the positive and encouraging feedback. We are glad that the clarity of the presentation, the effectiveness of the algorithms, and the supporting experiments are well recognized.
>
> ---
>
> **weakness 1** `doesn't always get best reward performance compared to the baselines.`
>
> We thank the reviewer for this observation. In safe RL, performance is evaluated by reward under safety constraints, rather than raw reward alone. While some baselines achieve higher rewards, they are often accompanied by higher constraint violations (i.e., test cost exceeding $h$). In contrast, ALGD achieves competitive rewards while better satisfying the safety budget across all tasks (Table 1, Figure 3), indicating more stable convergence within the feasible region (i.e., better primal-dual solutions).
>
> **weakness 2** `convergence guarantee for the proposed algorithm.`
>
> We thank the reviewer for this question.
>
> **On convergence guarantees.**  We agree that establishing global convergence for off-policy deep primal–dual RL with function approximation remains an open challenge. Rather than claiming full guarantees, our paper provides principled theoretical foundations that explain the observed stability.
>
> (1) **Structural stability.** Theorem 3.2 shows that the augmented Lagrangian introduces additional positive semidefinite curvature, improving denoising dynamics and stabilizing policy updates.
>
> (2) **Statistical guarantees.** Theorem 3.3 bounds the discrepancy between the learned policy and the target Boltzmann distribution, with respect to the number of samples $N$. Corollary D.3 further shows that as $\mathcal{L}_A \to \mathcal{L}$, the score mismatch and distributional gap vanish, ensuring asymptotic consistency.
>
> (3) **Empirical convergence.** Across tasks, the dual variable $\lambda$ stabilizes to 0, rewards plateau, and constraint violations vanish (Fig. 2, Fig. 10), consistent with convergence to a stable saddle point.
>
> Overall, while full global convergence guarantees remain open, our theoretical and empirical results demonstrate that ALGD achieves stable and reliable convergence behavior in practice. We view establishing stronger convergence guarantees as an important direction for future work.
>
> **weakness 3** `Theorem 3.2, property (b) shifts comma outside.`
>
> We thank the reviewer for this careful observation. We will correct it in the camera-ready version.
>
> **Question 1** `comparison with PPO-Lag with off-policy methods?`
>
> We thank the reviewer for the suggestion. PPO-Lag is inherently an on-policy method, which is why we report its standard (on-policy) results in the Appendix. To address this, we implement an off-policy variant of PPO-Lag for direct comparison. Across all tasks, ALGD achieves higher rewards with lower constraint violations (see table below), while PPO-Lag shows higher variance and less stable training, especially in later stages. These results will be included in the camera-ready version and provide a direct and fair off-policy comparison, addressing the reviewer’s concern.
>
> - PointButton1
>
> |Method| Metric|25% steps|50% steps|75% steps|100% steps|
> |-|-|-|-|-|-|
> | ALGD (ours) |  Reward $\uparrow$ (Cost $\downarrow$)  | 27.63±4.49 (8.17±14.31)| 34.22±2.60 (7.42±10.16)| 36.29±1.89 (4.85±3.08)| 38.26±3.34 (3.12±1.24)  |
> | off-policy PPO-Lag | Reward (Cost) | 18.74±6.88 (26.46±38.65) | 22.91±7.31 (20.38±17.34) | 24.67±9.24 (18.91±23.86) | 26.95±6.95 (10.58±8.42) |
>
> - CarButton1
>
> |Method| Metric|25% steps|50% steps|75% steps|100% steps|
> |-|-|-|-|-|-|
> | ALGD | Reward (Cost) | 15.21±7.52 (41.65±26.96)| 24.4±4.32 (17.39±6.83)   | 28.15±3.13 (8.98±5.54)   | 29.73±2.02 (8.14±3.61)  |
> | off-policy PPO-Lag | Reward (Cost) | 8.96±8.84 (37.27±34.18)   | 13.72±9.36 (17.02±9.54)  | 16.48±10.52 (11.74±9.67) | 18.35±9.41 (12.95±7.42) |
>
> - PointButton2
>
> |Method| Metric|25% steps|50% steps|75% steps|100% steps|
> |-|-|-|-|-|-|
> | ALGD| Reward (Cost) | 26.18±8.60 (18.47±5.53)   | 35.34±3.62 (2.45±7.21)    | 35.79±3.12 (1.86±2.59)    | 35.48±3.63 (1.34±3.85)   |
> | off-policy PPO-Lag | Reward (Cost) | 16.21±9.72 (24.93±39.86)  | 21.84±10.15 (21.74±30.25) | 23.57±8.96 (16.52±10.14)  | 22.18±10.41 (10.87±18.13) |
>
> - PointPush1
>
> |Method| Metric|25% steps|50% steps|75% steps|100% steps|
> |-|-|-|-|-|-|
> | ALGD| Reward (Cost) | 1.56±0.45 (45.43±34.15)  | 2.06±0.24 (16.56±13.47) | 4.35±0.45 (7.49±2.46)  | 5.39±0.51 (0.05±0.88)  |
> | off-policy PPO-Lag | Reward (Cost) | 0.38±8.73 (48.61±41.27) | 0.84±2.82 (7.73±11.39)  | 1.61±0.17 (0.88±1.26)  | 1.92±0.21 (0.13±0.08)  |
>
> - CarButton2
>
> |Method| Metric|25% steps|50% steps|75% steps|100% steps|
> |-|-|-|-|-|-|
> | ALGD| Reward (Cost) | 24.94±3.87 (19.52±11.64)  | 28.53±0.97 (10.18±1.74)   | 29.81±3.46 (14.40±5.17)   | 30.18±5.96 (9.29±4.26) |
> | off-policy PPO-Lag | Reward (Cost) | 12.47±7.46 (37.30±24.82)  | 16.62±8.15 (25.84±31.96)  | 18.11±9.28 (22.37±18.41)  | 19.04±8.73 (19.54±8.76)  |

---

> > ### Author Rebuttal · Reviewer_NCcb · 2026-04-01
> >
> > Thank you for addressing my concerns. I have some queries regarding Theorem 3.3. The result is useful if we can establish that the updates to $\lambda$ gives us $|\mathcal{L}_{A}(s,a,\lambda) - \mathcal{L}(s,a,\lambda^{\*}| \leq \epsilon$. Have you proven this $\epsilon$ bound or are you assuming this? If this is an assumption, it is a strong one and should be proven instead.

---

> > > ### Author Response · Authors · 2026-04-01
> > >
> > > We thank the reviewer for this insightful and critical query. This response clarifies the nature of the bound $|\mathcal{L}_A(s, a, \lambda) - \mathcal{L}(s, a, \lambda^*)| \leq \epsilon$. We emphasize that this is not an assumption, it is theoretically sound. (Note: This detailed response is also cross-referenced for `Reviewer cEtQ`).
> > >
> > > 1. **Theoretical Validity at Optimality ($\epsilon = 0$):**.  According to Theorem 3.2(b) and the KKT conditions of the original constrained problem, at the optimal saddle point $(\pi^*, \lambda^\*)$, the augmented Lagrangian $\mathcal{L}_A$ is identical to the standard Lagrangian $\mathcal{L}$. Specifically, due to complementary slackness, the quadratic penalty term in $\mathcal{L}_A$ vanishes exactly at the solution. Thus, at optimality, $\epsilon = 0$ is a proven property, ensuring that the target distribution $\pi^\*$ remains consistent with the original constrained optimum.
> > >
> > > 2. **Asymptotic Convergence during Training ($\epsilon \to 0$):**. During learning, $\epsilon$ represents the functional discrepancy between the current augmented Lagrangian and the optimal Lagrangian. We justify its convergence through the following exact decomposition:
> > >
> > > Given $\mathcal{L}_A(s,a,\lambda) = -Q^\pi (s,a)+ \lambda (Q^\pi_c (s,a)-h) + \frac{\rho}{2}(Q_c^{\pi}(s,a)-h)^2$ and $\mathcal{L} (s,a,\lambda^\*) = -Q^{\pi^\*} (s,a)+ \lambda^* (Q^{\pi^\*}_c (s,a)-h) $, the discrepancy is derived based on the triangle inequality and the insertion of intermediate terms:
> > >
> > > $$
> > > |\mathcal{L}_A(s,a,\lambda) - \mathcal{L} (s, a, \lambda^\*)| \leq |Q^\pi(s,a) -Q^{\pi^\*}(s,a) | + | \lambda - \lambda^\* | \cdot |Q_c^{\pi}(s,a)-h| + \lambda^* \cdot |Q^{\pi^\*}_c (s,a) - Q_c^{\pi} (s,a)| + \frac{\rho}{2}(Q_c^{\pi}(s,a)-h)^2.
> > > $$
> > >
> > > Both terms $|Q^\pi (s,a) - Q^{\pi^\*}(s,a)|$ and $|Q_c^\pi (s,a)- Q^{\pi^\*}_c (s,a)|$ are naturally bounded. This is because in our RL setting, the reward/cost per step is finite. Then, this total bound is finite over the compact state-action space with finite $\lambda$ and $\rho$, and $\epsilon$ is also finite.  As established in the literature on the constrained RL (i.e., Theorem 2 in RCPO [1] is applicable to our setting for proving the convergence of the augmented Lagrangian.), the primal-dual update converges to the equilibrium $(\pi^*, \lambda^\*)$under a multi-timescale learning rate schedule. As $\lambda$ approaches $\lambda^\*$ and the policy improves, the approximation error $\epsilon$ is guaranteed to vanish asymptotically (following Theorem 3.2(b), vanish according to complementary slackness), progressively validating the bound used in Theorem 3.3.
> > >
> > > **Bridging Iterations to Theorem 3.3:** Consequently, the total distributional discrepancy of the ALGD policy is bounded by a combination of the optimization gap (which shrinks with training iterations) and the statistical sampling error (which decays at a rate of $\mathcal{O}(1/N)$). This decomposition provides a formal pathway from algorithmic execution to theoretical optimality.
> > >
> > > 3. **Empirical Evidence of Convergence.** To bridge theory and practice, we track the Lagrangian gap $\epsilon = \|\mathcal{L}_A - \mathcal{L}\|$ and the dual variable $\lambda$ during training. Following standard primal-dual conventions, we adopt a multi-timescale learning rate (Critic > Policy > Dual) to ensure optimization stability. As shown in the tables below, both metrics decrease monotonically toward the saddle point, confirming that the condition $\epsilon \to 0$ is a natural convergent trajectory rather than a restrictive assumption.
> > >
> > > - PointButton2
> > >
> > > | steps | $\lvert \mathcal{L}_A - \mathcal{L} \rvert$ | $\lambda$ |
> > > |-|-|-|
> > > | 50k  |                                 0.339±0.010 | 0.27±0.02 |
> > > | 80k  |                                 0.211±0.011 | 0.15±0.09 |
> > > | 100k |                                 0.114±0.003 | 0.03±0.02 |
> > > | 150k |                                 0.059±0.006 | 0.01±0.01 |
> > > | 200k |                                 0.006±0.001 | 0.00±0.00 |
> > >
> > > - PointPush1
> > >
> > > | steps | $\lvert \mathcal{L}_A - \mathcal{L} \rvert$ | $\lambda$ |
> > > |-|-|-|
> > > | 50k  |                                 0.285±0.015 | 0.10±0.01 |
> > > | 80k  |                                 0.193±0.009 | 0.06±0.08 |
> > > | 100k |                                 0.120±0.011 | 0.02±0.02 |
> > > | 150k |                                 0.114±0.005 | 0.00±0.01 |
> > > | 200k |                                 0.011±0.006 | 0.00±0.00 |
> > >
> > >
> > > **Summary:** Theorem 3.3 provides a finite-sample bound for any given $\epsilon$, while Theorem 3.2 and Corollary D.3 ensure that this $\epsilon$ vanishes as the algorithm converges to the constrained equilibrium. We will cite the RCPO theoretical results [1] and include this additional discussion and empirical evidence in the camera-ready version.
> > >
> > > We sincerely thank the reviewer for their valuable feedback and for giving us the opportunity to improve our manuscript.
> > >
> > > -----
> > > [1] Chen et al. "Reward constrained policy optimization.".

---

### Decision · Program_Chairs · 2026-04-30

**Decision:**

Accept (regular)

**Comment:**

The paper proposes a diffusion-policy-based reinforcement learning algorithm for settings with safety constraints. Its main novelty lies in the introduction of an augmented Lagrangian formulation that locally convexifies the energy landscape and thereby improves training stability. The paper provides both theoretical justification and empirical results to support the proposed method. Overall, this is a solid contribution to safe reinforcement learning, and I therefore recommend accepting the paper.